# The Final Layer Holds the Key: A Unified and Efficient GNN Calibration Framework

## Abstract

Graph Neural Networks (GNNs) have demonstrated remarkable effectiveness on graph-based tasks. However, their predictive confidence is often miscalibrated, typically exhibiting *under-confidence*, which harms the reliability of their decisions. Existing calibration methods for GNNs normally introduce additional calibration components, which fail to capture the intrinsic relationship between the model and the prediction confidence, resulting in limited theoretical guarantees and increased computational overhead. To address this issue, we propose a simple yet efficient graph calibration method. We establish a unified theoretical framework revealing that model confidence is jointly governed by class-centroid-level and node-level calibration at the final layer. Based on this insight, we theoretically show that reducing the weight decay of the final-layer parameters alleviates GNN under-confidence by acting on the class-centroid level, while node-level calibration acts as a finer-grained complement to class-centroid level calibration, which encourages each test node to be closer to its predicted class centroid at the final-layer representations. Extensive experiments validate the superiority of our method.

## 1 Introduction

Graph Neural Networks (GNNs) have achieved significant success in recent years, becoming a powerful tool for learning from graph-structured data. Their ability to effectively model complex relationships between nodes and edges has led to widespread applications, including molecular biology (Cai et al., 2022; Wieder et al., 2020), recommendation systems (Fan et al., 2022; 2019), and knowledge graphs (Ye et al., 2022; Arora, 2020). Notably, even if the accuracy of GNNs meets high standards, the reliability of model outputs is critically important in real-world deployment. However, recent studies (Wang et al., 2021b; Hsu et al., 2022; Wang et al., 2022) have revealed that GNNs normally generate unreliable confidence, characterized by *under-confidence*—a sharp contrast to the over-confidence commonly observed in traditional Deep Neural Networks (DNNs). Consequently, traditional calibration methods cannot directly address the calibration issue of GNNs.

Current graph calibration methods can be roughly divided into two categories *i.e.,* regularization methods and post-hoc methods. Regularization methods (Wang et al., 2022; Stadler et al., 2021) introduce extra regularization terms specifically designed to calibrate GNNs during training. However, regularization methods were shown to struggle with effectively balancing accuracy and calibration (Hsu et al., 2022; Yang et al., 2024). Post-hoc methods (Wang et al., 2021b; Hsu et al., 2022; Tang et al., 2024) are proposed to tune the confidence after model training, preserving the model's accuracy while improving the calibration performance, thereby overcoming the above issue. Most existing post-hoc calibration methods for GNNs adopt a common framework, where an additional calibration GNN is trained on the validation set to learn suitable temperature scaling coefficients, which are then applied via temperature scaling (TS) (Guo et al., 2017) for calibration. The main differences among these methods lie in the inputs to the calibration GNN. For example, CaGCN (Wang et al., 2021b) takes the output probabilities of the base GNN; GATS (Hsu et al., 2022) incorporates neighborhood similarity and predictive distributions, as well as other related factors; and SimCalib (Tang et al., 2024) leverages node similarity and homophily.

Despite the effectiveness of existing post-hoc graph calibration methods, there are still two limitations that need to be addressed. First, previous studies focus on alleviating the confidence bias through external operations , which lacks an in-depth exploration of the intrinsic relationship between

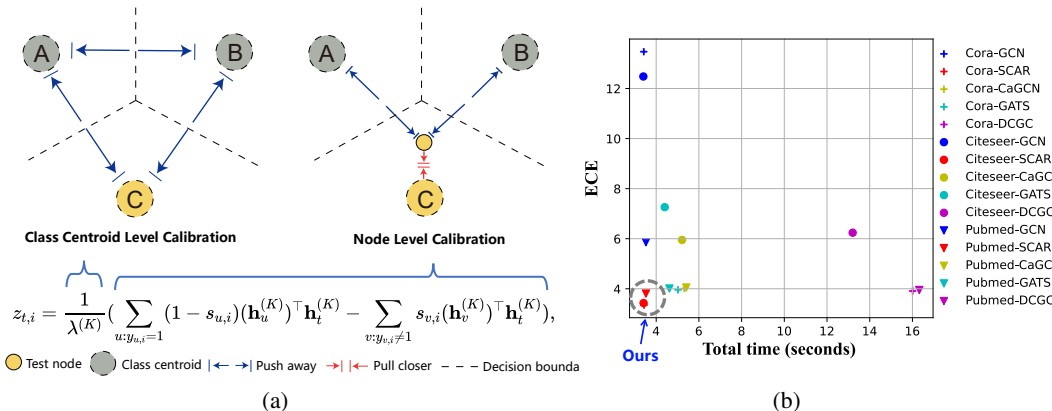

(a)                                      (b)

Figure 1: (a) An illustration of the proposed SCAR. The balls within the dotted box represent the class centroids, while the yellow dots inside the solid box represent the test nodes classified as class $C$. The output logits are determined by two components. The first is the distance between class centroids (*i.e.,* class-centroid-level calibration), which is controlled by the weight decay coefficient of the final layer. By reducing this coefficient, we push the class centroids away from each other, increasing the class separability. The second component is the distance from each test node to its corresponding class centroid (*i.e.,* node-level calibration). By reducing this distance as a post-hoc method, we further refine the confidence of each test node in a more fine-grained manner to alleviate under-confidence. (b) Scatter plot illustrating the trade-off between runtime and ECE (lower is better). Scatters closer to the lower-left corner exhibit superior performance.

the model itself and the under-confidence issue. Second, existing post-hoc calibration methods identify some potential factors (*i.e.,* the input of the calibration GNN) that are associated with model confidence, without uncovering how these factors affect it. Hence, they have to rely on additional neural networks and supervision from a validation set to learn the mapping in a black-box manner.

To address the above limitations, this paper conducts a theoretical analysis of the relationship between model parameter updates and prediction confidence, and proposes a novel method, **S**imple yet efficient graph **CA**lib**R**ation method (**SCAR**), as shown in Figure 1(a). To do this, we first reveal that the weight decay imposed on the final-layer parameters exacerbates the under-confidence issue of GNNs. To address this issue, we propose reducing the weight decay of the final-layer parameters to get a better-calibrated GNN. Moreover, our in-depth analysis shows that this adjustment is equivalent to enlarging the inter-class distance at the class-centroid level, thereby improving class separability. Since the confidence of each test node is closely related to its own representations, we introduce a node-level calibration strategy that encourages each test node to move closer to its predicted class centroid, serving as a fine-grained complement to class-centroid-level calibration. This method directly adjusts the representation of test nodes by reducing their distances to the centroids, without learning additional mapping functions, thereby offering a training-free and interpretable solution. Finally, we establish a unified theoretical framework demonstrating that model confidence is jointly governed by both class-centroid-level and node-level calibration, highlighting the completeness and coherence of our approach, as shown at the bottom of Figure 1(a). Based on this, the proposed SCAR achieves state-of-the-art performance in both effectiveness and efficiency, as shown in Figure 1(b).

Compared with previous graph calibration methods[1], our contributions can be summarized as follows:

- To the best of our knowledge, we provide the first work to theoretically reveal that the weight decay imposed on the final-layer parameters exacerbates the under-confidence of GNNs by affecting the model at the class-centroid level. We address this issue by reducing the weight decay of the final-layer parameters.

- We propose a node-level calibration as a training-free post-hoc method, serving as a fine-grained complement to class-centroid-level calibration and offering a principled foundation for future post-hoc confidence calibration approaches.

---

[1]The details of related work are summarized in the Appendix A.

- Theoretically demonstrate that model confidence is jointly governed by the proposed class-centroid-level and node-level calibration and empirically validate the effectiveness and efficiency of the proposed method across diverse settings, outperforming numerous state-of-the-art graph calibration methods.

## 2 PRELIMINARY

**Notations.** Given a graph $\mathcal{G} = (V, E, \mathbf{X}, \mathbf{Y})$, where $V$ is the node set and $E$ is the edge set. The original node representation is denoted by the feature matrix $\mathbf{X} \in \mathbb{R}^{n \times d}$ where $n$ is the number of nodes and $d$ is the number of features for each node. The label matrix is denoted by $\mathbf{Y} \in \mathbb{R}^{n \times c}$ with a total of $c$ classes. The sparse matrix $\mathbf{A} \in \mathbb{R}^{n \times n}$ is the adjacency matrix of $\mathcal{G}$. Let $\mathbf{D} = \text{diag}(d_1, d_2, \cdots, d_n)$ be the degree matrix, where $d_i = \sum_{j \in \mathcal{N}_i} a_{ij}$ is the degree of node $i$, the symmetric normalized adjacency matrix is represented as $\widehat{\mathbf{A}} = \widetilde{\mathbf{D}}^{-\frac{1}{2}} \widetilde{\mathbf{A}} \widetilde{\mathbf{D}}^{-\frac{1}{2}}$ where $\widetilde{\mathbf{A}} = \mathbf{A} + \mathbf{I}$, $\mathbf{I}$ is the identity matrix and $\widetilde{\mathbf{D}}$ is the degree matrix of $\widetilde{\mathbf{A}}$.

**Graph Neural Networks.** Graph Neural Networks are powerful models for processing graph data. Most GNNs follow the message-passing framework. Specifically, every node iteratively updates its representations by aggregating the representations of its neighbors. Formally, the $i$-th layer of a GNN can be expressed as:

$$\mathbf{f}_v^{(i)} = \text{COMB}(\mathbf{h}_v^{(i)}), \mathbf{h}_v^{(i)} = \text{AGG}(\{\mathbf{f}_u^{(i-1)} : u \in \mathcal{N}_v\})), \tag{1}$$

where $\mathbf{f}_v^{(i)}$ is the representation of node $v$ at the $i$-th layer, $\mathbf{h}_v^{(i)}$ is the representation after aggregation and $\mathcal{N}_v$ is a set of nodes adjacent to $v$. $\text{AGG}(\cdot)$ denotes a differentiable, permutation-invariant function (e.g., sum, mean), $\text{COMB}(\cdot)$ denotes a differentiable transformation function such as multi-layer perceptrons (MLPs). Different types of GNNs have different choices of $\text{COMB}(\cdot)$ and $\text{AGG}(\cdot)$. If there is no $\text{AGG}(\cdot)$ in Eq. (1) (*i.e.,* $\mathbf{h}_v^{(i)} = \mathbf{f}_v^{(i-1)}$), then it becomes a traditional MLP.

**Expected Calibration Error (ECE).** Let $\mathbf{s}_v$ be the prediction probabilities of node $v$ and the number of layers in the model is $K$, we have $\mathbf{s}_v = \text{softmax}(\mathbf{z}_v)$, where $\mathbf{z}_v$ is the logits of node $v$. The confidence for node $v$ is $p_v = \max_j s_{v,j}$. A model is said to be perfectly calibrated (Wang et al., 2021b) if its confidence scores exactly correspond to the actual probabilities. For example, with $0.8$ being the average confidence, $80\%$ of the predicted examples should be correct. Nevertheless, GNNs are often found to exhibit *under-confidence*, where the confidence scores are lower than the actual probability. For example, more than $80\%$ of predictions may be correct at an average confidence of $0.8$.

$$\text{PC: } \Pr[\hat{y}_v = y_v | p_v = p] = p, \forall p \in [0, 1], \quad \text{UC: } \Pr[\hat{y}_v = y_v | p_v = p] > p, \forall p \in [0, 1], \tag{2}$$

where **PC** and **UC** denote perfect calibration and under-confidence, respectively. The calibration quality can be quantified by the expected calibration error (ECE) (Naeini et al., 2015; Guo et al., 2017). It divides predictions into confidence intervals (bins, *i.e.,* $\text{B}_1, \ldots, \text{B}_M$) and calculates the accuracy and average predicted confidence within each bin. The ECE is then computed as the weighted average of the absolute differences between accuracy and confidence across all bins, with weights proportional to the number of predictions in each confidence bin. **A lower ECE means** that the confidence more closely matches its actual accuracy, thus indicating **better calibration.** Formally, the ECE can be defined as

$$\text{ECE} = \sum_{m=1}^{M} \frac{|B_m|}{|N_{\text{test}}|} |\text{acc}(B_m) - \text{conf}(B_m)| \quad s.t. \begin{cases} \text{acc}(B_m) = \frac{1}{|B_m|} \sum_{i \in B_m} \mathbf{1}(y_i = \hat{y}_i), \\ \text{conf}(B_m) = \frac{1}{|B_m|} \sum_{i \in B_m} p_i \end{cases}, \tag{3}$$

where $|N_{\text{test}}|$ is the number of test nodes and $\mathbf{1}(\cdot)$ is the indicator function.

## 3 THE PROPOSED METHOD

**Overview.** To mitigate the under-confidence of GNNs by refining its inherent causes, we first theoretically show that the weight decay imposed on the final layer intensifies under-confidence. Thus, simply reducing the final layer's weight decay leads to better-calibrated predictions. Moreover, the in-depth analysis shows that this adjustment increases the distance between class centroids by

directly operating at the class-centroid level. Given that a test node's confidence is also closely related to its own representation, we introduce a node-level calibration strategy as a fine-grained complement to class-centroid-level calibration. This strategy reduces the distance between each test node and its predicted class centroid in the final-layer representation space. Furthermore, our unified theoretical analysis reveals that model confidence is jointly governed by class-centroid-level and node-level calibration. An overview of the proposed method is presented in Figure 1(a).

## 3.1 CLASS-CENTROID LEVEL CALIBRATION

Previous calibration methods calibrate the confidence through external methods. For example, GCL (Wang et al., 2022) designed a loss function that constrains the model outputs to have low entropy probabilities (*i.e.,* high confidence). CaGCN (Wang et al., 2021b) training a calibration model on the validation set to scale the output probabilities. However, they do not facilitate an in-depth exploration of the intrinsic relationship between the model itself and under-confidence. To address this issue, we analyze the impact of final-layer parameters on confidence during model updating, leveraging this to guide the model to produce better-calibrated confidence scores.

Specifically, the most commonly used cross-entropy loss with weight decay for node $v$ are as follows:

$$\mathcal{L}_v = -\sum\nolimits_{i=1}^{c} y_{v,i} \log s_{v,i} + \sum\nolimits_{k}^{K} \frac{\lambda^{(k)}}{2} ||\mathbf{W}^{(k)}||_F^2, \tag{4}$$

where $\lambda^{(k)}$ is a regularization coefficient of layer $k$ that controls the strength of the weight decay.

Then we have the following theorem (the proof is provided in the Appendix B.1):

**Theorem 3.1.** *Given the learning rate (i.e., $\eta$) and the final-layer parameters (i.e., $\mathbf{W}^{(K)}$). For an arbitrary node $v$ in the training stage, its output probabilities on class $i$ (i.e., $s_{v,i}$) is updated by:*

$$s'_{v,i} = \frac{e^{b_{v,i}/\tau}}{e^{b_{v,i}/\tau} + \sum_{j\neq i}^{c} e^{b_{v,j}/\tau} \cdot \psi_{i,j}}, \quad s.t. \begin{cases} \psi_{i,j} = e^{\eta(s_{v,i}-y_{v,i}-s_{v,j}+y_{v,j})(\mathbf{h}_v^{(K)})^\top \mathbf{h}_v'^{(K)}} \\ \tau = \frac{1}{1-\eta\lambda^{(K)}} \\ b_{v,i} = (\mathbf{W}_{:,i}^{(K)})^\top \mathbf{h}_v'^{(K)} \end{cases} \tag{5}$$

*where $s'_{v,i}$ and $\mathbf{h}_v'^{(K)}$ are the GNN updated results of $s_{v,i}$ and $\mathbf{h}_v^{(K)}$ after the next epoch.*

From Theorem 3.1, we can observe that during the update of output probabilities, the weight decay of the final layer only affects $\tau$, and $\tau = \frac{1}{1-\eta\lambda^{(K)}} > 1$. Therefore, Theorem 3.1 reveals that applying weight decay to the final-layer parameters (*i.e.,* $\mathbf{W}^{(K)}$) is equivalent to adding a temperature coefficient greater than 1 to the output probability, which increases the entropy of output probabilities, thereby exacerbating the under-confidence of GNNs. Hence, reducing $\lambda^{(K)}$ can help mitigate the under-confidence issue specific to GNNs.

Although prior work (Guo et al., 2017) has observed that adjusting the global weight decay can influence model confidence, it often results in significant drops in accuracy, making it unsuitable as a practical calibration tool. To support this claim and highlight the strengths of our proposed method, we empirically compare the accuracy and ECE of global and final-layer weight decay in Figure 2. The experiment details can be found in the section 4. The results show that regulating global weight decay is far less effective than our method in terms of calibration. More importantly, it negatively impacts accuracy, making confidence calibration meaningless, as effective calibration relies on a well-maintained accuracy level. This highlights our work as the first to propose adjusting weight decay as a calibration method, supported by theoretical analysis.

**Analyzing the effect of $\lambda^{(K)}$.** Since reducing the final-layer weight decay (*i.e.,* $\lambda^{(K)}$) directly affects the parameter $\mathbf{W}^{(K)}$, this parameter plays a crucial role in generating confidence scores. To better understand its effect, we conduct an in-depth analysis of $\mathbf{W}^{(K)}$, which not only reveals that a smaller $\lambda^{(K)}$ increases the distance between class centroids (*i.e.,* **class-centroid-level calibration**), but also highlights the necessity and lays the foundation for the design of subsequent node-level calibration. Specifically, we can analyze $\mathbf{W}^{(K)}$ by the following theorem (the proof is listed in Appendix B.2):

**Theorem 3.2** (Closed-form solution for $\mathbf{W}^{(K)}$). *Given the objective function Eq. (4), the solution of $\mathbf{W}^{(K)}$ can be represent as:*

$$(\mathbf{W}_{:,i}^{(K)})^* = \frac{1}{\lambda^{(K)}} \left( \sum\nolimits_{u:y_{u,i}=1} (1-s_{u,i})\mathbf{h}_u^{(K)} - \sum\nolimits_{v:y_{v,i}\neq 1} s_{v,i}\mathbf{h}_v^{(K)} \right). \tag{6}$$

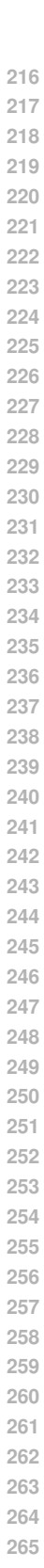
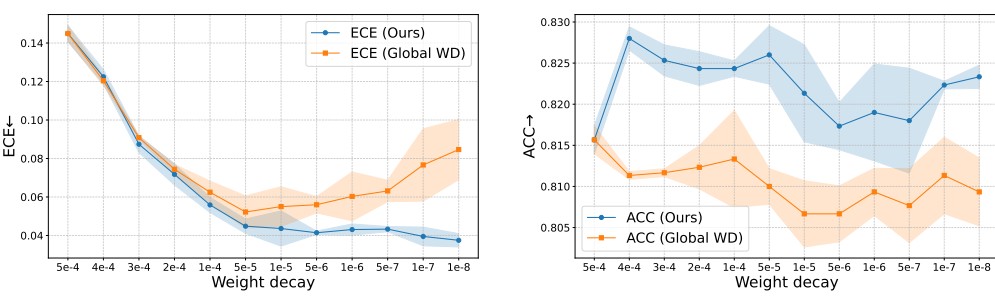

Figure 2: Impact of global vs. final-layer weight decay on Cora dataset (ECE(a), accuracy (b)).

From Theorem 3.2, we can see that each column of $(\mathbf{W}^{(K)})^*$ corresponds to a specific class. For instance, $(\mathbf{W}^{(K)}_{:,i})^*$ corresponds to class $i$, where $i \in [1, \ldots, c]$. Moreover, $(\mathbf{W}^{(K)}_{:,i})^*$ is the weighted sum of the final-layer node representations belonging to class $i$ in the training set, while subtracting the representations of nodes from other classes in the training set. Thus, every column of $(\mathbf{W}^{(K)})^*$ effectively represents the cluster centroids of each class in the training set. This also aligns with the deduction of the neural collapse theory (Papyan et al., 2020).

Building on the above observations, we can understand the effect of reducing the final-layer weight decay, according to the following proposition (the proof is provided in Appendix B.3):

**Proposition 3.3.** *Given two different coefficients* $\lambda_1, \lambda_2$ *of weight decay on the final-layer parameters (i.e.,* $\mathbf{W}^{(K)}$*), for any* $i, j \in [1, \ldots, c]$, $i \neq j$, *if* $\lambda_1 > \lambda_2$, *the following equation holds:*

$$||((\mathbf{W}^{(K)}_{:,i})^*|\lambda_1) - ((\mathbf{W}^{(K)}_{:,j})^*|\lambda_1)||_2^2 < ||((\mathbf{W}^{(K)}_{:,i})^*|\lambda_2) - ((\mathbf{W}^{(K)}_{:,j})^*|\lambda_2)||_2^2, \quad (7)$$

*where* $((\mathbf{W}^{(K)})^*|\lambda_1)$ *and* $((\mathbf{W}^{(K)})^*|\lambda_2)$ *denote the optimized parameters under weight decay coefficients* $\lambda_1$ *and* $\lambda_2$, *respectively.*

Proposition 3.3 indicates that **reducing the weight decay of** $\mathbf{W}^{(K)}$ **increases the distances between the class centroids of training nodes**, as illustrated in Figure 1 (a), thereby making the class centroids more clearly separable and ensuring that samples are more likely to fall into high-confidence regions.

### 3.2 NODE-LEVEL CALIBRATION

The above method performs calibration at the class-centroid level by controlling the distances between class centroids. Since the confidence of each test node is also closely related to its own representation, it is necessary to adjust each node's representation as a fine-grained complement to the class-centroid-level calibration, enabling more precise control over individual prediction confidence. Fortunately, based on the above theoretical analysis, we can easily derive the relationship between a test node's representation and its confidence. Specifically, given the closed-form solution of the final-layer parameters (*i.e.,* $(\mathbf{W}^{(K)})^*$) in Eq. (6), we can calculate the output logit for arbitrary test node $t$:

$$z_{t,i} = ((\mathbf{W}^{(K)}_{:,i})^*)^\top \mathbf{h}_t^{(K)} = \frac{1}{\lambda^{(K)}} \Big( \underbrace{\sum_{u:y_{u,i}=1} (1 - s_{u,i})(\mathbf{h}_u^{(K)})^\top \mathbf{h}_t^{(K)}}_{\text{intra-class similarity}} - \underbrace{\sum_{v:y_{v,i}\neq 1} s_{v,i}(\mathbf{h}_v^{(K)})^\top \mathbf{h}_t^{(K)}}_{\text{inter-class similarity}} \Big),$$

$$(8)$$

where $\mathbf{h}_t^{(K)}$ is the final-layer aggregated representation of $\mathbf{x}_t$. In Eq. (8), the first factor $\frac{1}{\lambda^{(K)}}$ has already been analyzed in the previous section. Therefore, we now focus on the second factor. This factor reveals that for an arbitrary test node $t$, the output logits for class $i$ (*i.e.,* $z_{t,i}$) are composed of two terms (*i.e.,* intra-class similarity and inter-class similarity). The first term calculates the similarity between the $i$-th class centroid (*i.e.,* $\sum_{u:y_{u,i}=1}(1 - s_{u,i})(\mathbf{h}_u^{(K)})^\top$) and the test node $t$ (*i.e.,* $\mathbf{h}_t^{(K)}$). The second term subtracts the similarity values between the test node $t$ and the other class centers, excluding the centroid of class $i$ (*i.e.,* $\sum_{v:y_{v,i}\neq 1} s_{v,i}(\mathbf{h}_v^{(K)})^\top$). Then we can conclude that in the final layer if the representation of a test node $t$ is closer to the $i$-th class centroid and farther from other

class centroids, the $i$-th element of the logits (*i.e.,* $z_{t,i}$) will obtain a large positive value; otherwise, it will obtain a large negative value. Obviously, when the logit corresponding to the predicted class is a very large positive value and the logits for the other classes are very large negative values, the softmax function produces a high-confidence prediction. This inspires us that **we can increase the confidence of GNN by shortening the distance between the test node representation and the corresponding classified class centroid of the training set in the final layer.**

Based on the above analysis, we design node-level calibration, a **post-hoc** and **training-free** method. Its schematic is provided in Appendix C.1. After training the GNN is completed, we can adjust the final-layer representation of the test node $t$ (*i.e.,* $\mathbf{h}_t^{(K)}$) using the following equation:

$$\mathbf{h}_t^{(K)} = \alpha \mathbf{W}_{:,i}^{(K)} + (1-\alpha)\mathbf{h}_t^{(K)}, \ \ s.t. \ \hat{\mathbf{y}}_{t,i} = 1 \tag{9}$$

where $\alpha \in [0,1]$ is a hyperparameter to control the similarity between node representation and its predicted class centroid, and $\hat{y}_t$ is the predicted label of node $t$. As we analyzed above, each column of $\mathbf{W}^{(K)}$ represents each class centroid. Therefore, Eq. (9) encourages each test node to move closer to its predicted class centroid. Meanwhile, the class-centroid-level calibration increases the distance between class centroids, which in turn pushes the test node $t$ farther away from other class centroids. This is fully consistent with the analysis of Eq. (8).

**Remark.** Eq. (8) provides a unified formulation that reveals model confidence is jointly determined by two multiplicative factors. As shown in Figure 1(a), the first factor corresponds to class-centroid-level calibration, which enhances global class separability; the second factor corresponds to node-level calibration, which adjusts each node's representation relative to its predicted class centroid. This factorized form reflects the complementary nature of the two calibration strategies and highlights the completeness of our proposed method.

**Analyzing GNN Bias in Node-Level Calibration.** In the proposed node-level calibration. the class-centroid (*i.e.,* $\mathbf{W}^{(K)}$) is calculated by the training nodes. Therefore, the distance between the test node and the training node is a key factor for the node-level calibration. However, within the GNN framework, the distance from the test node to the training node is biased. Specifically, test nodes that are closer to training nodes in graph structure tend to obtain more similar representations to them through the GNN's message-passing mechanism (Hsu et al., 2022; Cai & Wang, 2020; Rusch et al., 2023). Therefore, we further refine the proposed node-level calibration to mitigate the bias:

$$\begin{cases} \mathbf{h}_f^{(K)} = \alpha \mathbf{W}_{:,i}^{(K)} + (1-\alpha)\mathbf{h}_f^{(K)}, \ \ s.t. \ \hat{\mathbf{y}}_{f,i} = 1, f \in V_F \\ \mathbf{h}_s^{(K)} = \beta \mathbf{W}_{:,i}^{(K)} + (1-\beta)\mathbf{h}_s^{(K)}, \ \ s.t. \ \hat{\mathbf{y}}_{s,i} = 1, s \in V_S \end{cases}, \tag{10}$$

where $V_F$ denotes the first-order neighbors of the training node, and $V_S$ denotes its second-order and higher-order neighbors. The hyperparameters $\alpha, \beta \in [0,1], \beta > \alpha$ control the similarity between node representation and class centroid, and $\hat{y}_v$ is the predicted label of node $v$.

In Eq. (10), we implement two levels of instance-wise tuning. First, the representation of every test node will be close to the centroid of its predicted class and pushed away from other' class centroids. Second, nodes that are not adjacent to the training set will receive greater weights to move closer to their corresponding class centroids compared to nodes that are adjacent to the training set.

## 4 EXPERIMENTS

In this section, we conduct experiments on four public datasets to evaluate the proposed method in terms of different settings Details of experiments are shown in Appendix D, and additional experiments are shown in Appendix E. The code is released at Anonymous code link.

### 4.1 EXPERIMENTAL SETUP

**Datasets.** Following (Wang et al., 2021b), we evaluate on four commonly used citation networks: Cora (Sen et al., 2008), Citeseer (Sen et al., 2008), Pubmed (Sen et al., 2008), and CoraFull (Bojchevski & Günnemann, 2017). Results on more diverse datasets are provided in Appendix E.3.

**GNNs to be calibrated.** In our experiments, we use the two most classic GNNs (*i.e.,* GCN (Kipf & Welling, 2017) and GAT (Velickovic et al., 2018)) as backbones to be calibrated. For GCN and GAT,

we follow parameters suggested by (Kipf & Welling, 2017) and (Velickovic et al., 2018). Additional results on more base models are presented in Appendix E.4.

**Comparison Methods.** The comparison methods include two commonly used post-hoc calibration methods in general neural networks (*i.e.,* TS (Guo et al., 2017) and MS (Kull et al., 2019)), one regularization method specially designed for GNNs (*i.e.,* AU-LS (Wang et al., 2024)), and three SOTA post-hoc methods specially designed for GNNs (*i.e.,* CaGCN (Wang et al., 2021b), GATS (Hsu et al., 2022), DCGC (Yang et al., 2024), and GETS (Zhuang et al., 2025)).

**Evaluation Protocol.** We follow the evaluation in previous works (Wang et al., 2021b), adopting Expected Calibration Error (ECE) with 20 bins as the metric for calibration. Besides, since fine-tuning the hyper-parameter of the weight decay in the final layer, the predicted label may have slight changes, we also adopt classification accuracy as an evaluation metric. We report the average and standard deviation of 10 runs for each pair split of a dataset.

Table 1: ECE (%) with 20 bins on different models for citation networks, considering various numbers of labels per class (L/C). "-" denotes the result is not meaningful. *Uncal.* represents the uncalibrated model. The best results are highlighted in black.

| Datasets | L/C | GCN | | | | | | | | |
|---|---|---|---|---|---|---|---|---|---|---|
| | | Uncal. | TS | MS | CaGCN | GATS | AU-LS | DCGC | GETS | SCAR |
| Cora | 20 | $13.47_{\pm0.63}$ | $4.88_{\pm0.55}$ | $4.14_{\pm0.57}$ | $4.01_{\pm0.67}$ | $3.96_{\pm0.46}$ | $4.32_{\pm0.23}$ | $3.91_{\pm0.36}$ | $3.83_{\pm0.41}$ | $\mathbf{3.35_{\pm0.53}}$ |
| | 40 | $11.34_{\pm0.47}$ | $4.17_{\pm0.72}$ | $3.72_{\pm0.46}$ | $4.07_{\pm0.54}$ | $3.82_{\pm0.74}$ | $3.69_{\pm0.38}$ | $3.53_{\pm0.65}$ | $3.72_{\pm0.39}$ | $\mathbf{2.67_{\pm0.33}}$ |
| | 60 | $9.37_{\pm0.49}$ | $3.55_{\pm0.54}$ | $3.64_{\pm0.61}$ | $3.74_{\pm0.44}$ | $3.56_{\pm0.36}$ | $4.09_{\pm0.81}$ | $3.42_{\pm0.66}$ | $3.26_{\pm0.67}$ | $\mathbf{2.60_{\pm0.62}}$ |
| Citeseer | 20 | $12.48_{\pm0.71}$ | $6.41_{\pm0.87}$ | $6.44_{\pm0.37}$ | $5.95_{\pm0.72}$ | $7.26_{\pm0.63}$ | $6.69_{\pm1.76}$ | $6.24_{\pm0.41}$ | $6.72_{\pm0.78}$ | $\mathbf{3.43_{\pm0.58}}$ |
| | 40 | $9.57_{\pm0.77}$ | $6.01_{\pm0.42}$ | $5.38_{\pm0.57}$ | $5.45_{\pm0.55}$ | $6.82_{\pm0.25}$ | $7.56_{\pm2.19}$ | $6.16_{\pm0.52}$ | $6.01_{\pm0.63}$ | $\mathbf{4.17_{\pm0.45}}$ |
| | 60 | $8.06_{\pm0.64}$ | $5.59_{\pm0.50}$ | $5.21_{\pm0.64}$ | $5.46_{\pm0.34}$ | $8.11_{\pm0.37}$ | $5.74_{\pm0.99}$ | $7.43_{\pm0.61}$ | $7.84_{\pm0.66}$ | $\mathbf{4.72_{\pm0.69}}$ |
| Pubmed | 20 | $5.86_{\pm0.77}$ | $5.41_{\pm0.38}$ | $4.76_{\pm0.42}$ | $4.05_{\pm0.60}$ | $4.01_{\pm0.13}$ | $6.48_{\pm1.35}$ | $3.95_{\pm0.21}$ | $3.93_{\pm0.37}$ | $\mathbf{3.81_{\pm0.47}}$ |
| | 40 | $4.44_{\pm0.55}$ | $4.46_{\pm0.63}$ | $4.36_{\pm0.63}$ | $4.02_{\pm0.40}$ | $3.89_{\pm0.52}$ | $5.16_{\pm1.25}$ | $3.65_{\pm0.44}$ | $3.72_{\pm0.29}$ | $\mathbf{3.16_{\pm0.52}}$ |
| | 60 | $4.45_{\pm0.97}$ | $3.67_{\pm0.60}$ | $3.18_{\pm0.64}$ | $3.11_{\pm0.48}$ | $3.23_{\pm0.43}$ | $5.09_{\pm0.68}$ | $3.13_{\pm0.14}$ | $3.03_{\pm0.26}$ | $\mathbf{2.67_{\pm0.63}}$ |
| CoraFull | 20 | $19.86_{\pm0.61}$ | $10.31_{\pm0.61}$ | - | $7.76_{\pm0.64}$ | $7.39_{\pm0.75}$ | $7.37_{\pm0.89}$ | $7.36_{\pm0.84}$ | $7.01_{\pm0.85}$ | $\mathbf{6.96_{\pm0.48}}$ |
| | 40 | $23.21_{\pm0.54}$ | $11.17_{\pm0.65}$ | - | $7.01_{\pm0.39}$ | $6.97_{\pm0.40}$ | $5.72_{\pm0.67}$ | $6.84_{\pm0.52}$ | $5.98_{\pm0.74}$ | $\mathbf{5.61_{\pm0.43}}$ |
| | 60 | $23.37_{\pm0.40}$ | $9.81_{\pm0.38}$ | - | $7.68_{\pm0.34}$ | $6.65_{\pm1.67}$ | $7.28_{\pm0.96}$ | $6.59_{\pm1.01}$ | $6.83_{\pm0.41}$ | $\mathbf{6.31_{\pm0.69}}$ |

## 4.2 EFFECTIVENESS ANALYSIS

We first evaluate the effectiveness of the proposed method on the four citation datasets under different label rates. We are reporting the calibration results evaluated by ECE in Table 1 and Table 2. Obviously, our method achieves the best performance in calibration tasks. More surprisingly, the overall accuracy of our method has also improved. The specific results are in Appendix E.2. In addition, experiments on heterophilic and large-scale graphs are provided in Appendix E.3, while results on more base models are reported in Appendix E.4.

First, compared with the traditional calibration methods (*i.e.,* TS and MS), the proposed SCAR always outperforms them by large margins. For example, the proposed SCAR on average improves by 4.56%, compared to the best traditional calibration method (*i.e.,* TS, MS behaves badly on datasets with many classes, e.g., CoraFull) based on GCN and GAT. This demonstrates that calibration on graphs is more difficult and requires specialized methods.

Second, compared to graph calibration methods, the proposed SCAR achieves the best results, followed by CaGCN, GATS, AU-LS, DCGC, and GETS. For example, our method on average improves by 1.6%, compared to the most recent graph calibration method GETS, across all datasets and label rates, whether the base model is GCN or GAT. This can be attributed to the fact that the proposed SCAR calibrates the confidence of GNNs from the model itself, enabling the calibration process to better align with the model's inherent representation and learning dynamics.

## 4.3 ABLATION STUDY

The proposed SCAR consists of two components: class-centroid-level calibration ( **+ CLC** for short) and node-level calibration (**+ NLC** for short). To verify the effectiveness of each component in the proposed method, we evaluate the ECE of all variants using GCN and GAT backbones across all datasets with L/C = 20. The results are reported in Table 3.

Table 2: ECE (%) with 20 bins on different models for citation networks, considering various numbers of labels per class (L/C). "-" denotes the result is not meaningful. *Uncal.* represents the uncalibrated model. The best results are highlighted in black.

| Datasets | L/C | GAT | | | | | | | | |
|---|---|---|---|---|---|---|---|---|---|---|
| | | Uncal. | TS | MS | CaGCN | GATS | AU-LS | DCGC | GETS | SCAR |
| Cora | 20 | $15.58_{\pm0.89}$ | $7.17_{\pm0.98}$ | $5.44_{\pm0.94}$ | $4.50_{\pm0.56}$ | $4.25_{\pm0.29}$ | $5.51_{\pm0.83}$ | $4.10_{\pm0.53}$ | $4.53_{\pm0.48}$ | $\mathbf{3.52_{\pm0.74}}$ |
| | 40 | $13.40_{\pm0.54}$ | $4.85_{\pm0.77}$ | $4.91_{\pm0.60}$ | $3.65_{\pm0.56}$ | $3.68_{\pm0.67}$ | $3.79_{\pm0.58}$ | $4.28_{\pm0.42}$ | $4.01_{\pm0.68}$ | $\mathbf{3.52_{\pm0.74}}$ |
| | 60 | $12.01_{\pm0.33}$ | $3.93_{\pm0.61}$ | $4.11_{\pm0.53}$ | $3.13_{\pm0.32}$ | $2.76_{\pm0.31}$ | $3.44_{\pm0.95}$ | $2.64_{\pm0.63}$ | $3.16_{\pm0.58}$ | $\mathbf{2.59_{\pm0.43}}$ |
| Citeseer | 20 | $15.34_{\pm0.50}$ | $9.16_{\pm0.87}$ | $6.33_{\pm0.98}$ | $5.72_{\pm0.68}$ | $6.01_{\pm0.65}$ | $8.12_{\pm0.13}$ | $5.88_{\pm0.15}$ | $5.85_{\pm0.33}$ | $\mathbf{4.37_{\pm0.83}}$ |
| | 40 | $12.52_{\pm0.87}$ | $7.97_{\pm0.31}$ | $5.90_{\pm0.54}$ | $5.32_{\pm0.54}$ | $6.19_{\pm0.85}$ | $7.24_{\pm0.87}$ | $5.26_{\pm0.48}$ | $5.43_{\pm0.47}$ | $\mathbf{3.54_{\pm0.65}}$ |
| | 60 | $10.90_{\pm0.59}$ | $6.48_{\pm0.71}$ | $5.19_{\pm0.91}$ | $5.25_{\pm0.76}$ | $5.48_{\pm0.53}$ | $6.47_{\pm0.60}$ | $5.44_{\pm0.36}$ | $5.39_{\pm0.76}$ | $\mathbf{4.24_{\pm0.35}}$ |
| Pubmed | 20 | $8.35_{\pm0.31}$ | $6.56_{\pm0.46}$ | $5.01_{\pm0.37}$ | $3.56_{\pm0.63}$ | $4.68_{\pm1.68}$ | $3.58_{\pm0.47}$ | $\mathbf{3.52_{\pm0.21}}$ | $4.16_{\pm0.37}$ | $3.78_{\pm0.84}$ |
| | 40 | $8.69_{\pm0.46}$ | $6.58_{\pm0.65}$ | $5.39_{\pm0.60}$ | $3.08_{\pm0.54}$ | $3.60_{\pm0.26}$ | $4.26_{\pm0.81}$ | $3.54_{\pm0.58}$ | $3.34_{\pm0.44}$ | $\mathbf{3.02_{\pm0.30}}$ |
| | 60 | $9.93_{\pm0.41}$ | $6.69_{\pm0.63}$ | $5.39_{\pm0.60}$ | $3.08_{\pm0.54}$ | $2.94_{\pm0.72}$ | $3.97_{\pm0.52}$ | $2.90_{\pm0.98}$ | $2.96_{\pm0.61}$ | $\mathbf{2.80_{\pm0.85}}$ |
| CoraFull | 20 | $21.19_{\pm0.36}$ | $11.01_{\pm0.51}$ | - | $7.88_{\pm0.60}$ | $6.94_{\pm0.53}$ | $8.89_{\pm0.59}$ | $6.64_{\pm0.63}$ | $6.79_{\pm0.71}$ | $\mathbf{6.41_{\pm0.63}}$ |
| | 40 | $24.38_{\pm0.42}$ | $11.33_{\pm0.83}$ | - | $7.38_{\pm0.60}$ | $6.19_{\pm0.45}$ | $8.15_{\pm0.67}$ | $5.84_{\pm0.52}$ | $5.95_{\pm0.54}$ | $\mathbf{5.52_{\pm0.46}}$ |
| | 60 | $24.97_{\pm0.18}$ | $11.33_{\pm0.52}$ | - | $8.49_{\pm0.69}$ | $5.58_{\pm0.66}$ | $8.36_{\pm0.63}$ | $5.34_{\pm0.63}$ | $5.21_{\pm0.69}$ | $\mathbf{4.75_{\pm0.64}}$ |

According to Table 3, we have the following conclusions. First, our method with the complete components achieves the best performance. For example, our method on average improves by 0.8%, compared to the best variant (*i.e.,* **+ NLC**), indicating that all the components are necessary for our method. This is consistent with our analysis in Section 3. Those are the two proposed components that promote each other and complement each other. Second, our method significantly outperforms traditional TS in terms of any individual component's performance. For example, the individual **CLC** and **NLC** on average improve by 2.4% and 2.5% compared with TS. This indicates that the proposed method not only enhances the overall performance but also strengthens the contribution of each component. More importantly, previous post-hoc graph calibration methods were based on TS and needed to employ an extra calibration model to learn the appropriate temperature coefficients for every test node. The proposed NLC, like TS, is also a training-free post-hoc method, but its superior performance highlights its potential as a stronger foundation for subsequent post-hoc approaches.

Table 3: ECE (%) of each component on all datasets with L/C=20. The best results are highlighted in black.

| +CLC | +NLC | GCN | | | | GAT | | | |
|---|---|---|---|---|---|---|---|---|---|
| | | Cora | Citeseer | Pubmed | CoraFull | Cora | Citeseer | Pubmed | CoraFull |
| | TS | $4.88_{\pm0.55}$ | $6.41_{\pm0.87}$ | $5.41_{\pm0.38}$ | $10.13_{\pm0.61}$ | $7.17_{\pm0.98}$ | $9.16_{\pm0.87}$ | $6.56_{\pm0.46}$ | $11.01_{\pm0.51}$ |
| ✓ | | $3.72_{\pm0.53}$ | $4.92_{\pm0.79}$ | $4.01_{\pm0.47}$ | $7.09_{\pm0.52}$ | $3.64_{\pm0.63}$ | $5.98_{\pm0.62}$ | $3.99_{\pm0.53}$ | $6.89_{\pm0.83}$ |
| | ✓ | $4.08_{\pm0.62}$ | $3.63_{\pm0.63}$ | $3.88_{\pm0.44}$ | $9.23_{\pm0.55}$ | $4.18_{\pm0.72}$ | $4.96_{\pm0.66}$ | $4.13_{\pm0.42}$ | $7.12_{\pm0.45}$ |
| ✓ | ✓ | $\mathbf{3.35_{\pm0.65}}$ | $\mathbf{3.43_{\pm0.58}}$ | $\mathbf{3.78_{\pm0.53}}$ | $\mathbf{6.96_{\pm0.48}}$ | $\mathbf{3.52_{\pm0.74}}$ | $\mathbf{4.37_{\pm0.83}}$ | $\mathbf{3.78_{\pm0.84}}$ | $\mathbf{6.41_{\pm0.63}}$ |

## 4.4 VISUALIZATION ANALYSIS

In order to provide an intuitive and clear understanding of the effectiveness of the proposed model. We present visualizations of reliability diagrams and confidence distribution histograms with L/C = 20, across different datasets. The visualizations are shown in Figure 3.

In Figure 3 (Top), the first row represents the Reliability diagrams of GCN, and the second row represents the Reliability diagrams of the proposed SCAR. The x-axis divides the model's confidence into 20 equal intervals, while the y-axis represents the average accuracy for each interval. The gray area indicates the expected output values and the blue area represents the model's actual output values. The closer the bars are to the diagonal line (*i.e.,* $y = x$), the better calibrated the model. In all datasets, the uncalibrated GCN shows higher average accuracy than confidence in most bins, indicating under-confidence. Our proposed method effectively mitigates this under-confidence issue, aligning the model's confidence more closely with its accuracy.

In Figure 3 (Bottom), we visualize the confidence distribution of test nodes from different perspectives, where the x-axis represents the model's confidence and the y-axis represents the density at various

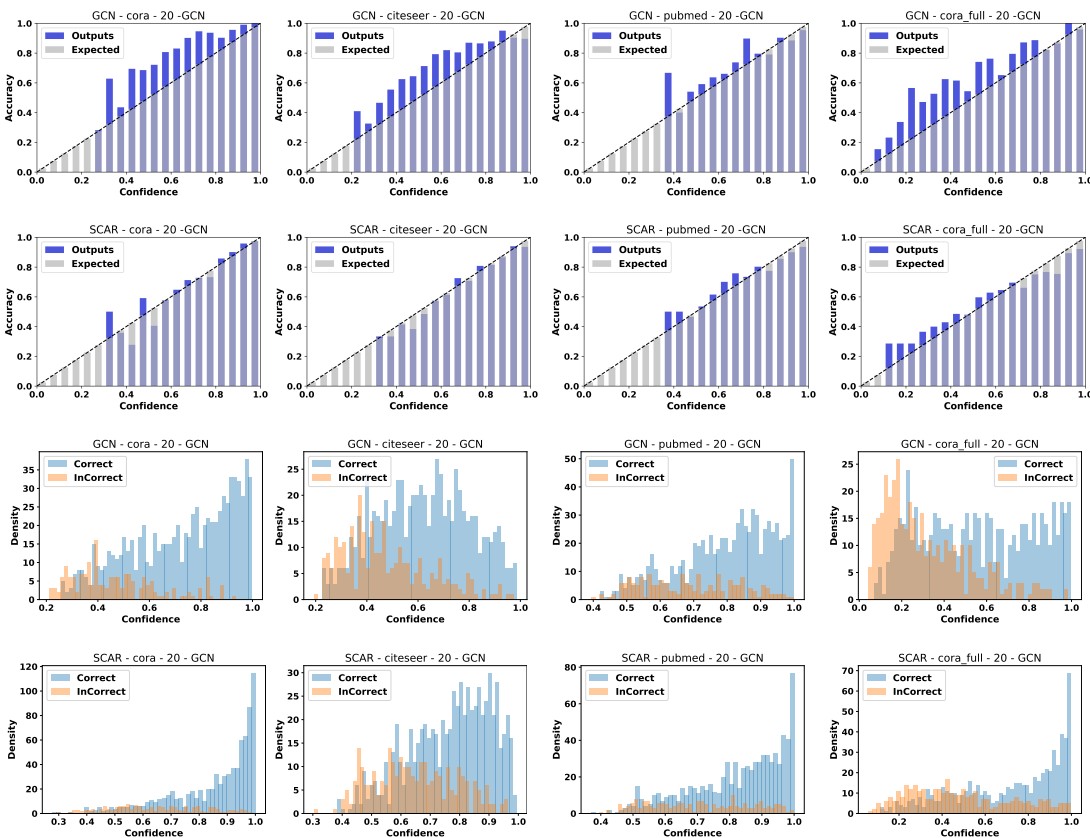

Figure 3: Visualization of Reliability diagrams (Top) and Confidence distribution histograms (Bottom) on Cora, Citeseer, Pubmed, and CoraFull datasets. The base model is GCN and L/C=20.

confidence levels. The blue represents the confidence distribution of correct predictions, while the yellow represents that of incorrect predictions. From this visualization, we observe that in the case of an uncalibrated GCN, many correctly predicted samples fall into lower confidence intervals. After applying the proposed method, most correctly predicted samples shift to higher confidence intervals, indicating that the model assigns greater confidence to its correct predictions. This shift underscores the superiority of the proposed method in improving the reliability of predictions.

## 5 CONCLUSION

In this paper, we conduct a comprehensive analysis of confidence calibration in GNNs. To do this, we first theoretically reveal that the weight decay imposed on the final-layer parameters aggravates the under-confidence of GNNs by shrinking the class centroids toward the origin, which reduces class separability. To mitigate this issue, we propose simply reducing the final-layer weight decay to enhance class-centroid distinction and improve confidence calibration at the class-centroid level. Meanwhile, the node-level calibration strategy is proposed as a fine-grained complement to class-centroid-level calibration, which encourages each test node to be closer to its predicted class centroid and far away from other class centroids in the final-layer representation space, thereby enhancing individual confidence calibration. Finally, we establish a unified theoretical framework showing that model confidence is jointly governed by both class-centroid-level and node-level calibration, which highlights the completeness and coherence of our method. Comprehensive experimental results demonstrate that the proposed method consistently outperforms state-of-the-art methods in terms of both effectiveness and efficiency on different datasets and settings.

## REPRODUCIBILITY STATEMENT

We have made extensive efforts to ensure the reproducibility of our work. The model architecture, training procedure, and hyperparameter settings are described in Appendix D.3. Complete proofs of the theoretical results are presented in Appendix B. The datasets used in our experiments are publicly available, and the preprocessing steps are explained in the Appendix D. Moreover, we provide an anonymous link to the source code in the supplementary materials to facilitate reproduction of our experiments (Anonymous code link).

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

## A  RELATED WORK

This section briefly reviews the topics related to this work, including graph neural networks, Uncertainty Quantification for GNNs and confidence calibration for GNNs.

### A.1  GRAPH NEURAL NETWORKS

Graph Neural Networks (GNNs) (Yang et al., 2023a; Kipf & Welling, 2017; Velickovic et al., 2018; Huang et al., 2023a; Yang et al., 2023b; Huang et al., 2023b; 2024; Mo et al., 2024b;a; Fang et al., 2022; 2025) have emerged as powerful tools for modeling and analyzing graph-structured data. Early approaches, such as the Graph Convolutional Network (GCN) (Kipf & Welling, 2017), leveraged spectral methods to define convolution operations on graphs, enabling effective semi-supervised learning on citation networks. Subsequent models, including GraphSAGE (Hamilton et al., 2017) and GAT (Velickovic et al., 2018), introduced sampling-based and attention mechanisms, respectively, to enhance scalability and expressiveness.

However, there are many problems in the basic research and practical application of GNN. For example, over-smoothing and robustness. Subsequent GNN works focus on solving the shortcomings of GNN. To address the challenges of over-smoothing and limited label utilization, recent advancements such as JK-Nets (Xu et al., 2018) and GCNII (Chen et al., 2020) proposed architectural modifications to preserve the previous layers' representation information for deep layer architecture and improve the performance of GNNs. For robustness issues, recent studies have also shown that GCNs are vulnerable to adversarial attacks. To solve this issue, recent studies proposed many robust GNNs. For example, Pro-GNN (Jin et al., 2020) de-noises graphs by constraining graph data to be low-rank, sparse, and feature smoothing. Mid-GCN (Huang et al., 2023a) find that mid-frequency signal in the spectral domain of the graph shows strong robustness, thus designing a mid-pass filtering GNN. Building on these advancements, an emerging line of research focuses on graph calibration, aiming to ensure that the model's confidence aligns with its predictive accuracy, which is crucial for trustworthy deployment in real-world applications.

### A.2  UNCERTAINTY QUANTIFICATION FOR GNNS

Recent advances have explored uncertainty estimation in graph neural networks (GNNs) from complementary perspectives. GCN-GEBM (Fuchsgruber et al., 2024) and GKDE (Zhao et al., 2020) model epistemic and aleatoric uncertainties from an energy-based viewpoint, where the uncertainty is quantified through structure-aware energy potentials or Dirichlet-based evidence distributions, improving robustness under semi-supervised and OOD settings. CF-GNN (Huang et al., 2023c) is the first to extend conformal prediction to graphs, providing theoretical coverage guarantees via calibrated prediction sets. However, these approaches mainly focus on uncertainty quantification or coverage assurance, and their effectiveness on confidence calibration remains underexplored.

### A.3  CONFIDENCE CALIBRATION FOR GNNS

Early research on confidence calibration primarily focused on the fields of computer vision and natural language processing (Guo et al., 2017; Niculescu-Mizil & Caruana, 2005; Rahimi et al., 2020; Wang et al., 2021a; Kumar et al., 2018; You et al., 2025; Yang et al., 2025). Guo et al. (2017) revealed that deep neural networks are often poorly calibrated and investigated the factors influencing calibration. Platt Scaling (Platt et al., 1999), originally proposed for binary classification models, adjusts raw model outputs into reliable class probability distributions by learning parameters optimized on a validation dataset. To calibrate on multi-class scenarios. Temperature Scaling (TS) (Guo et al., 2017) proposed to directly modify the model's output probabilities by introducing a temperature coefficient and fine-tuning on a validation set, which is the basis for much subsequent work.

Graph Calibration has emerged in recent years, and CaGCN (Wang et al., 2021b) was the first to discover that GNNs are under-confidence, which is very different from other modern neural networks, which are generally known to be over-confident. Therefore, calibration on GNNs needs more specific approaches. Current graph calibration methods can be classified into two categories, *i.e.,* regularization methods and post-hoc methods. Regularization methods produce calibration regularization terms when training GNNs. For example, GPN (Stadler et al., 2021) calibrates GNNs

by performing Bayesian posterior updates for predictions on interdependent nodes. GCL (Liu et al., 2022) achieves calibration by adding a minimal-entropy regularizer to the KL divergence. HyperU-GCN (Yang et al., 2022) further incorporates hyperparameter-level uncertainty into Auto-GNNs to improve calibration during training, but it requires additional optimization and model tuning. Post-hoc methods adjust the confidence when model training is completed, and current graph post-hoc methods follow a common paradigm that an additional calibration GNN is trained on the validation set to learn a suitable temperature coefficient mapping, which is then applied via TS for calibration test nodes. For example, CaGCN (Wang et al., 2021b) trains another calibration GNN model on the validation set by using the output probabilities as the input to learn the coefficient of temperature scaling to adjust the confidence, while GATS (Hsu et al., 2022) considers more factors as input, such as neighborhood similarities and homophily. SimCalib (Tang et al., 2024) relies on node similarity and homophily as input to the calibration GNN to learn the temperature coefficient. DCGNN (Fang et al., 2024) designs the self-loop same-class-neighbor ratio as the factor to input to the calibration GNN, improving calibration and discriminative ability. Otherwise, there is a data-centric method, DCGC (Yang et al., 2024), that can improve the existing post hoc method, which observed that a higher homophily ratio leads to better graph structure calibration. To leverage this, it learns a new homophily-based graph structure to enhance calibration. GETS (Zhuang et al., 2025) leverages a mixture-of-experts (MoE) architecture with diverse input transformations, and has demonstrated strong performance on graph calibration tasks. Although the above methods achieve excellent results, they were all designed by introducing extra components for calibration, while neglecting an in-depth exploration of the intrinsic relationship between the model itself and miss-calibration.

## B   PROOFS IN SECTION 3

### B.1   PROOF OF THEOREM 3.1

**Theorem B.1.** *Given the learning rate (i.e., $\eta$), the final-layer parameters (i.e., $\mathbf{W}^{(K)}$). For an arbitrary node $v$ in model training stage, its output probabilities on class $i$ (i.e., $s_{v,i}$) is updated by:*

$$s'_{v,i} = \frac{e^{b_{v,i}/\tau}}{e^{b_{v,i}/\tau} + \sum_{j \neq i}^c e^{b_{v,j}/\tau} \cdot \psi_{i,j}}$$

$$s.t., \quad \begin{cases} \psi_{i,j} = e^{\eta(s_{v,i}-y_{v,i}-s_{v,j}+y_{v,j})(\mathbf{h}_v^{(K)})^\top \mathbf{h}_v'^{(K)}} \\ \tau = \frac{1}{1-\eta\lambda^{(K)}} \\ b_{v,i} = (\mathbf{W}_{:,i}^{(K)})^\top \mathbf{h}_v'^{(K)} \end{cases}, \tag{11}$$

*where the value and representation $z'_{v,i}$ and $\mathbf{h}_v'^{(K)}$ are the updated results of $z_{v,i}$ and $\mathbf{h}_v^{(K)}$ after the next epoch.*

*Proof.* Given the CE and weight decay:

$$\mathcal{L}_v = -\sum_{i=1}^c y_{v,i} \log s_{v,i} + \sum_k^K \frac{\lambda^{(k)}}{2} ||\mathbf{W}^{(k)}||_F^2, \tag{12}$$

Then we can calculate the derivative of the final-layer parameters (*i.e.,* $\mathbf{W}^{(K)} \in \mathbb{R}^{d \times c}$):

$$\frac{\partial \mathcal{L}_v}{\partial \mathbf{W}_{:,i}^{(K)}} = \sum_{j=1}^c \left(\frac{\partial \mathcal{L}_v}{\partial s_{v,j}} \frac{\partial s_{v,j}}{\partial z_{v,i}}\right) \frac{\partial z_{v,i}}{\partial \mathbf{W}_{:,i}^{(K)}} + \frac{\partial \mathcal{L}_v}{\mathbf{W}_{:,i}^{(K)}} = (s_{v,i} - y_{v,i})\mathbf{h}_v^{(K)} + \lambda^{(K)}\mathbf{W}_{:,i}^{(K)}. \tag{13}$$

The details of derivation are listed in Appendix B.1.1. Then the final-layer parameter $\mathbf{W}^{(K)}$ is updated as:

$$\mathbf{W}_{:,i}'^{(K)} = (1 - \eta\lambda^{(K)})\mathbf{W}_{:,i}^{(K)} - \eta(s_{v,i} - y_{v,i})\mathbf{h}_v^{(K)}, \tag{14}$$

where $\mathbf{W}_{:,i}'^{(K)}$ is the next epoch of $\mathbf{W}_{:,i}^{(K)}$ and then the output logits $z_{v,i}$ rely on $\mathbf{W}^{(K)}$ is updated as:

$$\begin{aligned} z'_{v,i} &= (1 - \eta\lambda^{(K)})(\mathbf{h}_v'^{(l)})^\top \mathbf{W}_{:,i}'^{(K)} \\ &= (1 - \eta\lambda^{(K)})(\mathbf{h}_v'^{(l)})^\top \mathbf{W}_{:,i}^{(K)} - \eta(s_{v,i} - y_{v,i})(\mathbf{h}_v'^{(l)})^\top \mathbf{h}_v^{(l)}, \end{aligned} \tag{15}$$

where $\mathbf{h}_v'^{(l)}$ and $z_{v,i}'$ is the next epoch of $\mathbf{h}_v^{(l)}$ and $z_{v,i}$, respectively. Furthermore, the output probabilities can be obtained:

$$
\begin{aligned}
s_{v,i} &= \frac{e^{z_{v,i}'}}{e^{z_{v,i}'} + \sum_{k \neq j}^{c} e^{z_{v,k}'}} \\
&= \frac{e^{(1-\eta\lambda^{(K)})(\mathbf{h}_v'^{(l)})^\top \mathbf{W}_{:,i}^{(K)} - \eta(s_{v,i}-y_{v,i})(\mathbf{h}_v'^{(l)})^\top \mathbf{h}_v^{(l)}}}{e^{(1-\eta\lambda^{(K)})(\mathbf{h}_v'^{(l)})^\top \mathbf{W}_{:,i}'^{(K)} - \eta(s_{v,i}-y_{v,i})(\mathbf{h}_v'^{(l)})^\top \mathbf{h}_v^{(l)}} + \sum_{j \neq i}^{c} e^{(1-\eta\lambda^{(K)})(\mathbf{h}_v'^{(l)})^\top \mathbf{W}_{:,i}^{(K)} - \eta(s_{v,j}-y_{v,j})(\mathbf{h}_v'^{(l)})^\top \mathbf{h}_v^{(l)}}}.
\end{aligned}
\tag{16}
$$

We let $\tau = \frac{1}{1-\eta\lambda^{(K)}}$ and $b_i = (\mathbf{h}_v'^{(l)})^\top \mathbf{W}_{:,i}^{(K)}$, then we have:

$$
\begin{aligned}
s_{v,i} &= \frac{e^{b_i/\tau} / e^{\eta(s_{v,i}-y_{v,i})(\mathbf{h}_v'^{(l)})^T \mathbf{h}_v^{(l)}}}{e^{b_i/\tau} / e^{\eta(s_{v,i}-y_{v,i})(\mathbf{h}_v'^{(l)})^T \mathbf{h}_v^{(l)}} + \sum_{j \neq i}^{c} e^{b_i/\tau} / e^{\eta(s_{v,i}-y_{v,i})(\mathbf{h}_v'^{(l)})^T \mathbf{h}_v^{(l)}}}. \\
&= \frac{e^{b_i/\tau}}{e^{b_i/\tau} + \sum_{j \neq i}^{c} e^{b_i/\tau} \cdot e^{\eta(s_{v,i}-y_{v,i}-s_{v,j}+y_{v,j})(\mathbf{h}_v^{(K)})^\top \mathbf{h}_v'^{(K)}}}.
\end{aligned}
\tag{17}
$$

We let $\psi_{i,j} = e^{\eta(s_{v,i}-y_{v,i}-s_{v,j}+y_{v,j})(\mathbf{h}_v^{(K)})^\top \mathbf{h}_v'^{(K)}}$, thus, we can get the finally form:

$$
s_{v,i}' = \frac{e^{b_{v,i}/\tau}}{e^{b_{v,i}/\tau} + \sum_{j \neq i}^{c} e^{b_{v,j}/\tau} \cdot \psi_{i,j}}
\tag{18}
$$

$\square$

### B.1.1 DERIVATION OF EQ.(13)

*Proof.*

$$
\frac{\partial \mathcal{L}_v}{\partial \mathbf{W}_{:,i}^{(K)}} = \sum_{j=1}^{c} \left( \frac{\partial \mathcal{L}_v}{\partial s_{v,j}} \frac{\partial s_{v,j}}{\partial z_{v,i}} \right) \frac{\partial z_{v,i}}{\partial \mathbf{W}_{:,i}^{(K)}} + \frac{\partial \mathcal{L}_v}{\mathbf{W}_{:,i}^{(K)}}
\tag{19}
$$

We can calculate the derivatives of each part separately.

$$
\frac{\partial \mathcal{L}}{\partial s_{v,j}} = \frac{\partial \sum_{i=1}^{c} y_{v,i} \log s_{v,i} + \lambda^{(K)} \sum_{k}^{K} ||\mathbf{W}^{(k)}||_F^2}{\partial s_{v,j}} = -\frac{y_{v,j}}{s_{v,j}}.
\tag{20}
$$

For $\frac{\partial s_{v,j}}{\partial z_{v,i}}$, the forward equation is softmax function:

$$
s_{v,j} = \frac{e^{z_{v,j}}}{e^{z_{v,j}} + \sum_{k \neq j}^{c} e^{z_{v,k}}}.
\tag{21}
$$

There are two cases of derivation here. In the first case, when $j = i$, the independent variable appears in both the numerator and the denominator, we have

$$
\begin{aligned}
\frac{\partial \frac{e^{z_{v,i}}}{\sum_{k}^{c} e^{z_{v,k}}}}{\partial z_{v,i}} &= -\frac{e^{z_{v,i}} \cdot e^{z_{v,i}}}{(\sum_{k=1}^{c} e^{z_{v,k}})^2} + \frac{e^{z_{v,i}}}{\sum_{k=1}^{c} e^{z_{v,k}}} \\
&= \frac{e^{z_{v,i}}}{\sum_{k=1}^{c} e^{z_{v,k}}} \left(1 - \frac{e^{z_{v,i}}}{\sum_{k=1}^{c} e^{z_{v,k}}}\right) \\
&= s_{v,i}(1 - s_{v,i}).
\end{aligned}
\tag{22}
$$

When $j \neq i$, the independent variable appears only in the denominator, it is easy to obtain:

$$
\frac{\partial \frac{e^{z_{v,j}}}{\sum_{k}^{c} e^{z_{v,k}}}}{\partial z_{v,i}} = -\frac{e^{z_{v,j}}}{\sum_{k}^{c} e^{z_{v,k}}} \cdot \frac{e^{z_{v,i}}}{\sum_{k}^{c} e^{z_{v,k}}} = -s_{v,k} s_{v,i}.
\tag{23}
$$

Finally, we have

$$\frac{\partial \mathcal{L}_v}{\partial \mathbf{W}_{:,i}^{(K)}} = \left(\sum_{k\neq i}^{c} \frac{y_{v,k}}{s_{v,k}}.s_{v,k}.s_{v,i} - \frac{y_{v,i}}{s_{v,i}}.s_{v,i}.(1-s_{v,i})\right)\frac{\partial s_{v,i}}{\mathbf{W}_{:,i}^{(K)}} + \frac{\partial \lambda^{(K)}\sum_k^K ||\mathbf{W}^{(k)}||_F^2}{\mathbf{W}_{:,i}^{(K)}}. \quad (24)$$

$$= (s_{v,i} - y_{v,i})\mathbf{h}_v^{(K)} + \lambda^{(K)}\mathbf{W}_{:,i}^{(K)}$$

$\square$

## B.2 PROOF FOR THEOREM 3.2

**Theorem B.2.** *(Closed-form Solution for $\mathbf{W}^{(K)}$) Given the objective function Eq. (4), the solution of $\mathbf{W}^{(K)}$ can be represent as:*

$$(\mathbf{W}_{:,i}^{(K)})^* = \frac{1}{\lambda^{(K)}}\left(\sum_{u:y_{u,i}=1}(1-s_{u,i})\mathbf{h}_u^{(K)} - \sum_{v:y_{v,i}\neq 1}s_{v,i}\mathbf{h}_v^{(K)}\right) \quad (25)$$

*Proof.* For the one training node $v$, we can obtain the derivative of $\mathbf{W}_{:,i}^{(K)}$ from the section B.1:

$$\frac{\partial \mathcal{L}_v}{\partial \mathbf{W}_{:,i}^{(K)}} = (s_{v,i} - y_{v,i})\mathbf{h}_v^{(K)} + \lambda^{(K)}\mathbf{W}_{:,i}^{(K)}. \quad (26)$$

Let Eq. (26) equal to 0, we can obtain the closed-form solution $(\mathbf{W}_{:,i}^{(K)})^*$ of Eq. (4), *i.e.,*

$$(\mathbf{W}_{:,i}^{(K)})^* = -\frac{1}{\lambda^{(K)}}(s_{v,i} - y_{v,i})\mathbf{h}_v^{(K)} \quad (27)$$

We can extend it to the case of multiple training nodes as follows:

$$(\mathbf{W}_{:,i}^{(K)})^* = -\sum_{u\in\mathcal{T}} \frac{1}{\lambda^{(K)}}(s_{u,i} - y_{u,i})\mathbf{h}_u^{(K)}$$

$$= \frac{1}{\lambda^{(K)}}\left(\sum_{u:y_{u,i}=1}(1-s_{u,i})\mathbf{h}_u^{(K)} - \sum_{v:y_{v,i}\neq 1}s_{v,i}\mathbf{h}_v^{(K)}\right) \quad (28)$$

$\square$

## B.3 PROOF FOR PROPOSITION 3.3

**Proposition B.3.** *Given two different coefficients $\lambda_1, \lambda_2$ of weight decay on the final-layer parameters (i.e., $\mathbf{W}^{(K)}$), for any $i, j \in [1, \ldots, c]$, $i \neq j$, if $\lambda_1 > \lambda_2$, the following equation holds:*

$$||((\mathbf{W}_{:,i}^{(K)})^*|\lambda_1) - ((\mathbf{W}_{:,j}^{(K)})^*|\lambda_1)||_2^2 < ||((\mathbf{W}_{:,i}^{(K)})^*|\lambda_2) - ((\mathbf{W}_{:,j}^{(K)})^*|\lambda_2)||_2^2, \quad (29)$$

*where $((\mathbf{W}^{(K)})^*|\lambda_1)$ and $((\mathbf{W}^{(K)})^*|\lambda_2)$ are the parameters updated under the weight decay with coefficients $\lambda_1$ and $\lambda_2$, respectively.*

*Proof.* The closed-from solution of $\mathbf{W}^{(K)}$ can be obtained from Theorem 3.2:

$$((\mathbf{W}^{(K)})^*|\lambda) = \frac{1}{\lambda}\left(\sum_{u:y_{u,i}=1}(1-s_{u,i})\mathbf{h}_u^{(K)} - \sum_{v:y_{v,i}\neq 1}s_{v,i}\mathbf{h}_v^{(K)}\right) \quad (30)$$

Then we have

$$||((\mathbf{W}_{:,i}'^{(K)})^*|\lambda_1) - ((\mathbf{W}_{:,j}'^{(K)})^*|\lambda_1)||_F^2 = ||\frac{1}{\lambda_1}\left(\sum_{u:y_{u,i}=1}(1-s_{u,i})\mathbf{h}_u^{(K)} - \sum_{v:y_{v,i}\neq 1}s_{v,i}\mathbf{h}_v^{(K)}\right)||_F^2$$

$$= \frac{1}{\lambda_1^2}||\sum_{u:y_{u,i}=1}(1-s_{u,i})\mathbf{h}_u^{(K)} - \sum_{v:y_{v,i}\neq 1}s_{v,i}\mathbf{h}_v^{(K)}||_F^2. \quad (31)$$

The same as $||((\mathbf{W}'^{(K)}_{:,i})^*|\lambda_2) - ((\mathbf{W}'^{(K)}_{:,j})^*|\lambda_2)||^2_F$:

$$||((\mathbf{W}'^{(K)}_{:,i})^*|\lambda_2) - ((\mathbf{W}'^{(K)}_{:,j})^*|\lambda_2)||^2_F = \frac{1}{\lambda_2^2}||\sum_{u:y_{u,i}=1}(1-s_{u,i})\mathbf{h}_u^{(K)} - \sum_{v:y_{v,i}\neq 1}s_{v,i}\mathbf{h}_v^{(K)}||^2_F. \quad (32)$$

Therefore, we can have:

$$||((\mathbf{W}'^{(K)}_{:,i})^*|\lambda_1) - ((\mathbf{W}'^{(K)}_{:,j})^*|\lambda_1)||^2_F - ||((\mathbf{W}'^{(K)}_{:,i})^*|\lambda_2) - ((\mathbf{W}'^{(K)}_{:,j})^*|\lambda_2)||^2_F$$

$$= \frac{1}{\lambda_1^2}||\sum_{u:y_{u,i}=1}(1-s_{u,i})\mathbf{h}_u^{(K)} - \sum_{v:y_{v,i}\neq 1}s_{v,i}\mathbf{h}_v^{(K)}||^2_F - \frac{1}{\lambda_2^2}||\sum_{u:y_{u,i}=1}(1-s_{u,i})\mathbf{h}_u^{(K)} - \sum_{v:y_{v,i}\neq 1}s_{v,i}\mathbf{h}_v^{(K)}||^2_F$$

$$= (\frac{1}{\lambda_1^2} - \frac{1}{\lambda_2^2})||\sum_{u:y_{u,i}=1}(1-s_{u,i})\mathbf{h}_u^{(K)} - \sum_{v:y_{v,i}\neq 1}s_{v,i}\mathbf{h}_v^{(K)}||^2_F$$

$$< 0.$$

$$(33)$$

Thus, we obtain $||((\mathbf{W}'^{(K)}_{:,i})^*|\lambda_1) - ((\mathbf{W}'^{(K)}_{:,j})^*|\lambda_1)||^2_F < ||((\mathbf{W}'^{(K)}_{:,i})^*|\lambda_2) - ((\mathbf{W}'^{(K)}_{:,j})^*|\lambda_2)||^2_F.$

$\square$

# C ADDITIONAL METHODOLOGICAL DETAILS

## C.1 SCHEMATIC OF NODE-LEVEL CALIBRATION

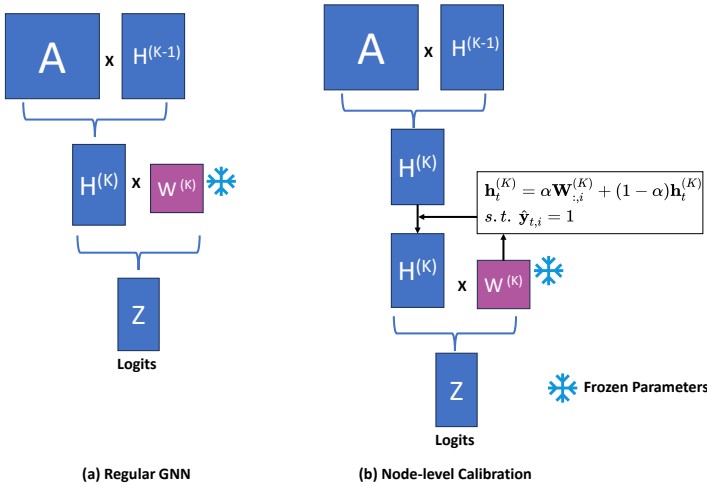

Figure 4: Illustration of the proposed node-level calibration in the final layer during test time.

The Schematic of Node-Level Calibration is shown in Figure 4. The proposed node-level calibration introduces only a lightweight adjustment to the final-layer representation $\mathbf{H}^{(K)}$, guided by the final-layer parameters $\mathbf{W}^{(K)}$ and predicted labels.

# D EXPERIMENTAL SETTINGS

This section provides detailed experimental settings in Section 4, including the description of all datasets in Section D.1, summarization of all comparison methods in Section D.2, model architectures and settings in Section D.3, and the evaluation protocol in Section D.4.

## D.1 DATASETS

We follow the previous works (Wang et al., 2021b) choose the commonly used Cora (Sen et al., 2008), Citeseer (Sen et al., 2008), Pubmed (Sen et al., 2008), and CoraFull (Bojchevski & Günnemann,

2017) for evaluation, where re composed of papers as nodes and their relationships such as citation relationships, and common authoring. Node feature is a one-hot vector that indicates whether a word is present in that paper. Words with a frequency of less than 10 are removed. We choose 500 nodes for validation and 1000 for testing and select three label rates for the training set (*i.e.,* 20, 40, and 60 labeled nodes per class). The details of these datasets are summarized in Table 4. All experiments setting are performed following the official code (Wang et al., 2021b).

Table 4: The statistics of the datasets

| Datasets | Nodes | Edges | Train | Valid | Test Nodes | Features | Classes |
|---|---|---|---|---|---|---|---|
| Cora | 2,708 | 5,429 | 140/280/420 | 500 | 1000 | 1,433 | 7 |
| Citeseer | 3,327 | 4,732 | 120/124/360 | 500 | 1,000 | 3,703 | 6 |
| Pubmed | 19,717 | 44,338 | 60/120/360 | 500 | 1,000 | 500 | 3 |
| CoraFull | 19,793 | 126,842 | 1400/2800/4200 | 500 | 1000 | 8,710 | 70 |

## D.2   COMPARISON METHODS

This study focuses on the classical semi-supervised node classification task. To evaluate model calibration, we follow the most authoritative work (Wang et al., 2021b) that considers two widely-used GNN architectures: GCN (Kipf & Welling, 2017) and GAT (Velickovic et al., 2018).

GCN is the most traditional undirected graph neural network, which updates node representations by aggregating neighboring formations. GAT is a graph neural network model that extends the self-attention mechanism to the graph, and updates nodes' representation by attention score to aggregate neighbors.

We compare several post-hoc calibration methods commonly applied to neural networks, including Temperature Scaling (TS) (Guo et al., 2017) and Matrix Scaling (MS) (Kull et al., 2019). Furthermore, the state-of-the-art graph calibration models are also compared:

- **CaGCN** (Wang et al., 2021b), the first calibration method designed specifically for GNNs, which employs a calibration model to scale the confidence of every node after training the mode.

- **GATS** (Hsu et al., 2022) is the same as CaGCN which both learn instance-wise temperatures based on a heuristic formula.

- **AU-LS** (Wang et al., 2024) is a regularization method for calibrating GNNs, utilizing a reverse label smoothing objective function to enhance the confidence of GNN predictions. We employ the 2-layer GNN variant, as the accuracy of the 3-layer GNN is significantly lower, making further discussion unnecessary. Additionally, the accuracy of the 2-layer AULS variants is also reported in Table E.2.

- **DCGC** (Yang et al., 2024) is a data-centric calibration method, that learns a homophily graph for better calibration. In the experimental results, we report the results of the best DCGC variant GATS-DCGC.

## D.3   MODEL ARCHITECTURES AND SETTINGS

In our experiments, we follow (Wang et al., 2021b), adopting a two-layer configuration for GCN and GAT, with hidden layer dimensions selected from $\{8, 16, 64\}$. All parameters are optimized using the Adam optimizer (Kingma & Ba, 2015) with the learning rate selected from $\{0.01, 0.015, 0.02\}$, dropout selected from $\{0.5, 0.6\}$, and the weight decay for all layers is set to 0.0005. In our proposed model, the weight decay of the final-layer parameter $\mathbf{W}^{(K)}$ is set to smaller than 0.0005 (used for the other layers' parameters). the hyper-parameters $\alpha, \beta$ are selected from $(0.0005, 0.000005]$ and $\alpha < \beta$. Table 5 and Table 6 describe the detailed settings and architecture for our experimental setups with SCAR based on GCN and GAT backbones, respectively.

Table 5: Settings for the proposed SCAR based on GCN.

| Datasets | L/C | Lr | Weight decay | Hidden units | Dropout | Weight decay of $\mathbf{W}^{(K)}$ | $\alpha$ | $\beta$ |
|---|---|---|---|---|---|---|---|---|
| Cora | 20 | 0.015 | 5e-4 | 64 | 0.6 | 1e-4 | 2e-4 | 2e-3 |
| | 40 | 0.015 | 5e-4 | 64 | 0.6 | 1e-4 | 1e-4 | 1e-3 |
| | 60 | 0.015 | 5e-4 | 64 | 0.6 | 1e-4 | 1e-4 | 1e-3 |
| Citeseer | 20 | 0.01 | 5e-4 | 64 | 0.5 | 4.5e-4 | 5e-5 | 5e-3 |
| | 40 | 0.01 | 5e-4 | 64 | 0.5 | 4.5e-4 | 5e-5 | 5e-3 |
| | 60 | 0.01 | 5e-4 | 64 | 0.5 | 4e-4 | 5e-4 | 2e-3 |
| Pubmed | 20 | 0.02 | 5e-4 | 16 | 0.5 | 4e-4 | 5e-8 | 5e-6 |
| | 40 | 0.02 | 5e-4 | 16 | 0.5 | 4e-4 | 5e-7 | 5e-5 |
| | 60 | 0.02 | 5e-4 | 16 | 0.5 | 4e-4 | 5e-7 | 5e-5 |
| CoraFull | 20 | 0.01 | 5e-4 | 64 | 0.5 | 2e-6 | 3e-5 | 5e-4 |
| | 40 | 0.01 | 5e-4 | 64 | 0.5 | 1e-4 | 3e-4 | 5e-3 |
| | 60 | 0.01 | 5e-4 | 64 | 0.5 | 2e-6 | 3e-5 | 5e-4 |

Table 6: Settings for the proposed SCAR based on GAT.

| Datasets | L/C | Lr | Weight decay | Hidden units | Dropout | Weight decay of $\mathbf{W}^{(K)}$ | $\alpha$ | $\beta$ |
|---|---|---|---|---|---|---|---|---|
| Cora | 20 | 0.01 | 5e-4 | 8 | 0.5 | 2e-5 | 5e-6 | 4e-4 |
| | 40 | 0.01 | 5e-4 | 8 | 0.5 | 2e-5 | 5e-6 | 4e-4 |
| | 60 | 0.01 | 5e-4 | 8 | 0.5 | 2e-5 | 5e-6 | 4e-4 |
| Citeseer | 20 | 0.01 | 5e-4 | 8 | 0.6 | 3e-4 | 5e-4 | 5e-3 |
| | 40 | 0.01 | 5e-4 | 8 | 0.6 | 3e-4 | 5e-4 | 5e-3 |
| | 60 | 0.01 | 5e-4 | 8 | 0.6 | 3e-4 | 5e-4 | 5e-3 |
| Pubmed | 20 | 0.01 | 5e-4 | 8 | 0.6 | 3e-4 | 5e-6 | 5e-5 |
| | 40 | 0.01 | 5e-4 | 6 | 0.5 | 3e-4 | 5e-7 | 5e-5 |
| | 60 | 0.01 | 5e-4 | 8 | 0.5 | 3e-4 | 5e-7 | 5e-6 |
| CoraFull | 20 | 0.01 | 5e-4 | 8 | 0.6 | 2e-6 | 3e-5 | 5e-4 |
| | 40 | 0.01 | 5e-4 | 8 | 0.6 | 2e-6 | 3e-5 | 5e-4 |
| | 60 | 0.01 | 5e-4 | 8 | 0.6 | 2e-6 | 3e-5 | 5e-4 |

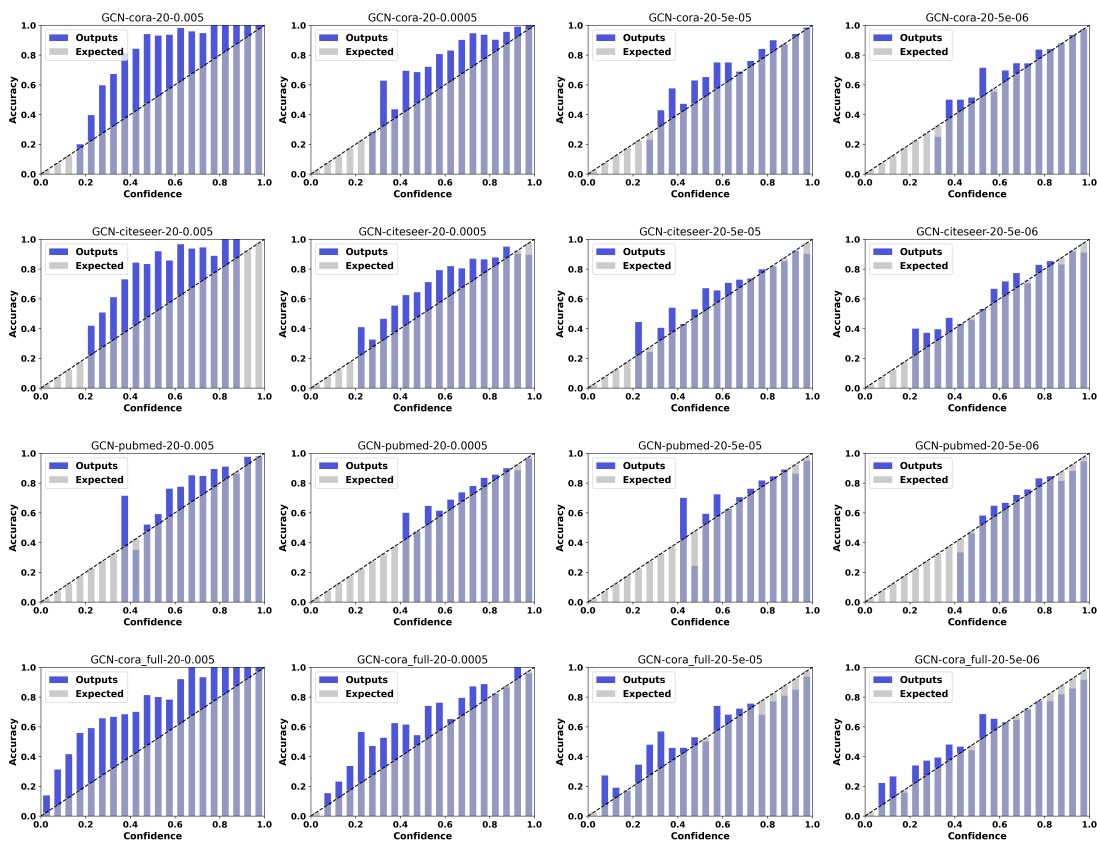

Figure 5: Reliability diagrams under different values of $\lambda^{(K)}$ (0.005, 0.0005, 0.00005, 0.000005) on the Cora, Citeseer, Pubmed, and CoraFull datasets. The base model is GCN and L/C=20.

### D.4 EVALUATION PROTOCOL

We follow the evaluation in previous works (Wang et al., 2021b), adopting Expected Calibration Error (ECE) with 20 bins as the metric for calibration. Besides, since fine-tuning the hyperparameter of the weight decay in the final layer, the predicted label may have slight changes, we also adopt classification accuracy as an evaluation metric. We report the average and standard deviation of 10 runs for each pair split of a dataset.

### D.5 COMPUTING RESOURCE DETAILS

All experiments were implemented in PyTorch and conducted on a server with 8 NVIDIA GeForce 4090 (24 GB memory each). Almost every experiment can be done on an individual 4090, and the training time of all comparison methods as well as our method, is less than 1 hour.

## E ADDITIONAL EXPERIMENTS

### E.1 ANALYSIS OF HYPER-PARAMETERS

In the proposed method, SCAR, we theoretically prove that the weight decay of the final layer leads to exacerbating the under-confidence of GNNs, thus we propose to reduce the final-layer weight decay to avoid the under-confidence phenomenon. Moreover, we employ the non-negative parameters (*i.e.,* $\alpha$ and $\beta$) to control the similarity between cluster centroid and test nodes, where $\alpha$ controls the test nodes adjacent to the training nodes, and $\beta$ controls the remaining test nodes.

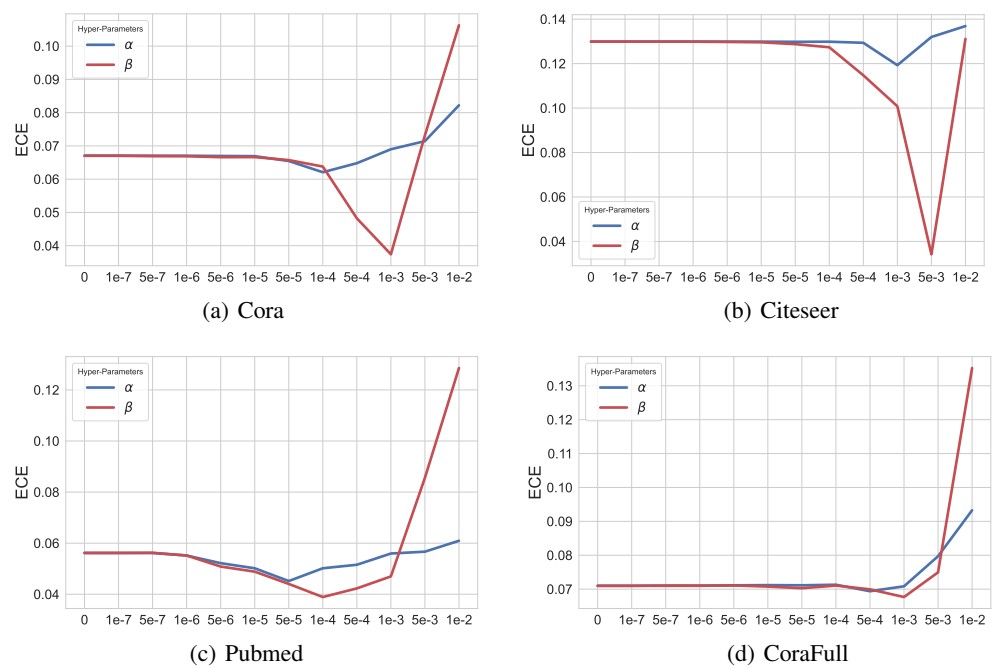

Figure 6: Visualization of the impact of defferent $\alpha$ and $\beta$ on ECE and red line represnet $\beta$, the blue line represent $\alpha$. The base model is GCN and L/C=20.

To investigate the impact of weight decay of the final layer, we conduct node classification on all datasets with L/C=20 by varying the value of the final-layer weight decay in the range of [0.005, 0.0005, 0.00005, 0.000005] (from left to right) and report the results in Figure 5. As shown in Figure 5, it is clearly observed that as the weight decay of the final layer decreases, the model transitions from being under-confident to confident, and its calibration performance improves progressively. This further validates the results of our Theorem 3.1 and highlights the importance of adjusting the weight decay of the final layer for improving confidence. For the selection of the final-layer weight decay, based on this observation, we recommend a binary search strategy, allowing us to efficiently determine the optimal weight decay value.

To investigate the impact of $\alpha$ and $\beta$ with different settings, we conduct the node classification on all datasets with L/C=20 by fixing a hyper-parameter and varying the value of another hyper-parameter (*i.e.,* both $\alpha$ and $\beta$) in the range of [0, 0.001]. Note that we have chosen a suitable weight decay parameter for $\mathbf{W}^{(K)}$. The results are reported in Figure 6. From Figure 6, we have the following observations. First, adjusting $\beta$ (*i.e.,* the red line) always obtains a lower ECE than $\alpha$ (*i.e.,* the blue line). This indicates that improving the similarity between test nodes disconnected from training nodes and their predicted class centroid is more effective than for connected test nodes. Second, the optimal value of $\beta$ is always higher than $\alpha$. This indicates that the test nodes disconnected from training nodes need a larger weight to move closer to the class centroid. The above observations are consistent with our analysis that test nodes farther from the training nodes are more under-confident. For the selection of $\alpha$ and $\beta$, Figure 6 already uggests that $\beta$ should be greater than $\alpha$. Leveraging this insight, we employed a partial grid search (i.e., searching only within the triangular region where $\beta > \alpha$), effectively reducing the search space by half while maintaining thorough exploration of meaningful parameter settings.

## E.2 ACCURACY ANALYSIS

The proposed SCAR framework comprises two components: class-centroid-level calibration and node-level calibration. Notably, since node-level calibration is a post-hoc method, which does not alter the model's predictions, while only class-centroid-level calibration (reducing the weight decay of the final layer) influences the predictions. Although adjusting the weight decay of the final layer is

typically considered part of standard hyperparameter tuning, we conducted experiments to evaluate its impact on accuracy compared with the uncalibrated model and regularization method (*i.e.,* AU-LS). The results are shown in Table 7.

From Table 7, we can observe that across all datasets and L/C configurations, SCAR consistently outperforms the uncalibrated models and regularization model (*i.e.,* AU-LS) for both GCN and GAT backbones. This demonstrates that reducing the weight decay of the final layer not only achieves a good calibration of the GNNs' confidence but also improves their prediction accuracy. It is worth noting that, especially in the CoraFull dataset, the proposed SCAR on average improves by 2.46 % and 5.7 %, compared to GCN and GAT.

Table 7: Accuracy of node classification (%), considering various numbers of labels per class (L/C). *Uncal.* represents the uncalibrated model. The best results are highlighted in black.

| Datasets | L/C | GCN | | | GAT | | |
|---|---|---|---|---|---|---|---|
| | | Uncal. | AU-LS | SCAR | Uncal. | AU-LS | SCAR |
| Cora | 20 | $81.53_{\pm 0.32}$ | $79.58_{\pm 0.14}$ | $\mathbf{82.01}_{\pm \mathbf{0.21}}$ | $82.56_{\pm 0.31}$ | $80.53_{\pm 1.05}$ | $\mathbf{83.45}_{\pm \mathbf{0.29}}$ |
| | 40 | $\mathbf{83.04}_{\pm \mathbf{0.24}}$ | $82.90_{\pm 0.69}$ | $82.70_{\pm 0.16}$ | $\mathbf{83.56}_{\pm \mathbf{0.25}}$ | $82.28_{\pm 0.99}$ | $83.35_{\pm 0.42}$ |
| | 60 | $84.35_{\pm 0.35}$ | $84.03_{\pm 0.80}$ | $\mathbf{84.49}_{\pm \mathbf{0.37}}$ | $85.06_{\pm 0.31}$ | $84.55_{\pm 0.61}$ | $\mathbf{85.07}_{\pm \mathbf{0.53}}$ |
| Citeseer | 20 | $\mathbf{71.71}_{\pm \mathbf{0.32}}$ | $65.65_{\pm 2.91}$ | $71.61_{\pm 0.45}$ | $71.47_{\pm 0.88}$ | $68.57_{\pm 1.01}$ | $\mathbf{71.51}_{\pm \mathbf{0.60}}$ |
| | 40 | $72.18_{\pm 0.32}$ | $70.71_{\pm 1.62}$ | $\mathbf{72.43}_{\pm \mathbf{0.41}}$ | $72.31_{\pm 0.56}$ | $71.55_{\pm 0.47}$ | $\mathbf{72.37}_{\pm \mathbf{0.31}}$ |
| | 60 | $72.98_{\pm 0.25}$ | $72.91_{\pm 0.49}$ | $\mathbf{73.01}_{\pm \mathbf{0.50}}$ | $72.91_{\pm 0.69}$ | $73.15_{\pm 0.38}$ | $\mathbf{73.23}_{\pm \mathbf{0.51}}$ |
| Pubmed | 20 | $79.53_{\pm 0.13}$ | $76.74_{\pm 2.61}$ | $\mathbf{79.60}_{\pm \mathbf{0.40}}$ | $\mathbf{78.88}_{\pm \mathbf{0.31}}$ | $77.92_{\pm 0.56}$ | $78.62_{\pm 0.36}$ |
| | 40 | $80.57_{\pm 0.34}$ | $78.59_{\pm 0.68}$ | $\mathbf{80.84}_{\pm \mathbf{0.23}}$ | $\mathbf{80.64}_{\pm \mathbf{0.43}}$ | $79.26_{\pm 0.30}$ | $80.21_{\pm 0.57}$ |
| | 60 | $83.11_{\pm 0.24}$ | $80.51_{\pm 0.51}$ | $\mathbf{83.24}_{\pm \mathbf{0.54}}$ | $82.27_{\pm 0.30}$ | $80.79_{\pm 0.36}$ | $82.07_{\pm 0.37}$ |
| CoraFull | 20 | $62.03_{\pm 0.27}$ | $61.62_{\pm 0.62}$ | $\mathbf{64.17}_{\pm \mathbf{0.52}}$ | $58.87_{\pm 0.35}$ | $54.71_{\pm 0.43}$ | $\mathbf{62.91}_{\pm \mathbf{0.32}}$ |
| | 40 | $65.02_{\pm 0.32}$ | $62.31_{\pm 0.67}$ | $\mathbf{67.21}_{\pm \mathbf{0.46}}$ | $60.08_{\pm 0.30}$ | $58.39_{\pm 0.48}$ | $\mathbf{66.17}_{\pm \mathbf{0.48}}$ |
| | 60 | $66.71_{\pm 0.33}$ | $64.12_{\pm 0.83}$ | $\mathbf{69.77}_{\pm \mathbf{0.31}}$ | $61.39_{\pm 0.27}$ | $59.37_{\pm 0.52}$ | $\mathbf{68.38}_{\pm \mathbf{0.37}}$ |

### E.3 Validation on Heterophilic Dataset

Table 8: ECE (%) with 20 bins on different models for heterophilic datasets, where the Arxiv-year dataset is also included as a representative large-scale graph. The best results are highlighted in black.

| Datasets | Chameleon | Squirrel | Arxiv-year |
|---|---|---|---|
| GCN | $13.82_{\pm 0.75}$ | $11.67_{\pm 0.18}$ | $13.34_{\pm 0.72}$ |
| SCAR-GCN | $\mathbf{7.30}_{\pm \mathbf{1.48}}$ | $\mathbf{7.25}_{\pm \mathbf{0.35}}$ | $\mathbf{6.03}_{\pm \mathbf{0.38}}$ |
| GAT | $14.11_{\pm 0.23}$ | $11.13_{\pm 0.83}$ | $12.33_{\pm 0.87}$ |
| SCAR-GAT | $\mathbf{7.93}_{\pm \mathbf{0.13}}$ | $\mathbf{8.12}_{\pm \mathbf{0.53}}$ | $\mathbf{5.94}_{\pm \mathbf{0.75}}$ |

To comprehensively evaluate the effectiveness and generalization of the proposed method, we conduct experiments on a wide range of graph datasets. In addition to commonly used homophilic benchmarks, we further include heterophilic graphs (*i.e.,* Chameleon (Pei et al., 2020), Squirrel (Pei et al., 2020), and Arxiv-year Platonov et al. (2023), wherein Arxiv-year is also a large-scale graphs) to verify the adaptability of our approach under diverse structural and label distribution settings. The experiment results are shown in Table 8.

From Table 8, we can observe that across all heterophilic datasets, the proposed SCAR consistently improves the ECE by a large margin, especially on the large-scale Arxiv-year dataset. Specifically, on Arxiv-year, SCAR reduces the ECE by 54.8% and 51.8% when applied to GCN and GAT, respectively. This demonstrates the generality of our method across diverse types of graph datasets.

Table 9: ECE (%) with 20 bins on different models for deferent base model (*i.e.,* GCNII and FAGCN). The best results are highlighted in black.

| ECE | Cora | Citeseer | Pubmed | Chameleon | Squirrel |
|---|---|---|---|---|---|
| GraphSAGE | $9.89_{\pm0.78}$ | $10.08_{\pm0.88}$ | $7.38_{\pm0.65}$ | $13.27_{\pm0.81}$ | $11.14_{\pm0.48}$ |
| SCAR-GraphSAGE | $\mathbf{4.88}_{\pm0.71}$ | $\mathbf{4.32}_{\pm0.73}$ | $\mathbf{5.86}_{\pm0.46}$ | $\mathbf{6.74}_{\pm0.86}$ | $\mathbf{7.17}_{\pm0.51}$ |
| FAGCN | $15.94_{\pm0.17}$ | $20.56_{\pm4.5}$ | $5.83_{\pm0.63}$ | $15.83_{\pm0.63}$ | $10.36_{\pm0.24}$ |
| SCAR-FAGCN | $\mathbf{4.57}_{\pm0.60}$ | $\mathbf{4.92}_{\pm1.4}$ | $\mathbf{3.74}_{\pm0.71}$ | $\mathbf{6.96}_{\pm0.84}$ | $\mathbf{7.85}_{\pm0.43}$ |
| GCNII | $35.51_{\pm0.56}$ | $35.30_{\pm1.11}$ | $5.72_{\pm1.77}$ | $18.13_{\pm0.26}$ | $17.79_{\pm0.61}$ |
| SCAR-GCNII | $\mathbf{9.81}_{\pm0.48}$ | $\mathbf{8.74}_{\pm0.51}$ | $\mathbf{3.98}_{\pm0.88}$ | $\mathbf{7.03}_{\pm0.35}$ | $\mathbf{8.93}_{\pm0.37}$ |

### E.4 VALIDATION ON MORE BASE MODEL

Beyond GCN and GAT, which are the most commonly used backbones in semi-supervised node classification, we further evaluate SCAR on more powerful architectures, including inductive GNN (*i.e.,* GraphSAGE (Hamilton et al., 2017)), heterophilic GNN (i.e., FAGCN (Bo et al., 2021)), and DeepGNN (i.e., GCNII (Chen et al., 2020)). These models incorporate advanced mechanisms such as residual connections and adaptive filtering, enabling them to capture more complex patterns in graph data. We conduct experiments on five datasets: Cora, Citeseer, and Pubmed (with 20 labeled nodes per class), as well as Chameleon and Squirrel, which are known to be heterophilous. The results are summarized in the Table 9.

The results of the Table 9 demonstrate that SCAR consistently improves calibration (ECE) across these models. This is because our method does not rely on any assumptions regarding graph homophily or specific message-passing mechanisms.

### E.5 ANALYZING THE TRADE-OFF BETWEEN EFFICIENCY AND EFFECTIVENESS

The proposed SCAR consists of two components. The first component involves a simple adjustment of the hyperparameters (i.e., class-centroid-level calibration), which does not introduce any additional complexity or runtime overhead. The second component, node-level calibration, modifies the final-layer representation of the test nodes during model inference, as described in Eq. (10). The computational complexity of Eq. (10) is minimal, as the overhead caused by calculating the weighted sum once time is negligible. To evaluate the model's actual runtime performance, we present the running time and ECE of each model based on GCN across different datasets with L/C=20 in Figure 7. Note that a smaller ECE value indicates better calibration. Therefore, models positioned closer to the lower-left corner of the plot achieve a better trade-off between efficiency and effectiveness.

From Figure 7, we have the following observations: First, existing graph calibration methods fail to achieve a good trade-off between effectiveness and efficiency. For example, the trade-off results of the most effective baseline (*i.e.,* DCGC), on all datasets (represented by the pink points) are clustered in the lower-right corner. This indicates that, while DCGC achieves well-calibrated results, it incurs significant additional time overhead. In contrast, baselines with slightly worse ECE performance, such as GATS, demonstrate improvements in runtime efficiency. Second, the proposed SCAR (represented by the red points), on all datasets is clustered in the lower-left corner. In more detail, on the time axis, the proposed SCAR and its backbone model GCN are at the same level, indicating almost no time overhead generated by the proposed SCAR. On the ECE axis, the proposed method is unparalleled, achieving a distinct advantage over the baselines. This demonstrates that the proposed method has achieved the best trade-off between efficiency and effectiveness. This is attributed to the fact that the proposed calibration method is designed based on the mechanism of generating confidence within the GNNs. As a result, it relies solely on minor modifications to the model itself, without the need for any external components, thus introducing almost no additional time overhead.

### E.6 ABLATION STUDY ON NODE-LEVEL CALIBRATION

To further validate the refined node-level calibration in Eq. (10), we conduct an ablation study comparing (i) the full version that uses both parameters $\alpha$ and $\beta$, which explicitly accounts for the

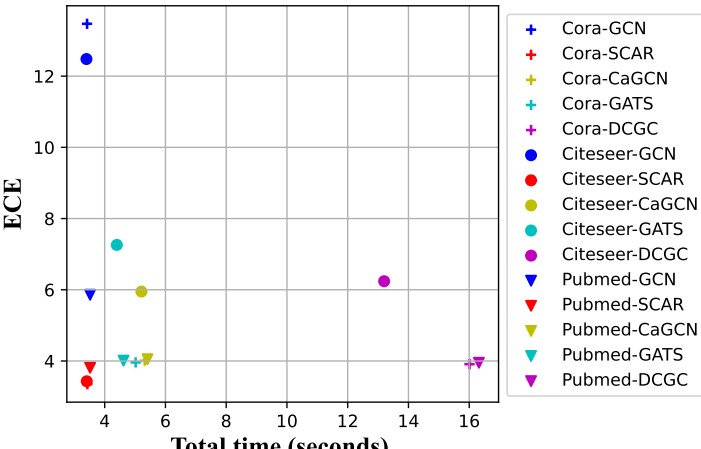

Figure 7: Scatter plot showing the relationship between model runtime and ECE, where the x-axis represents the runtime of different models on different datasets and the y-axis represents their corresponding ECE (%).

structural bias of GNNs, and (ii) a simplified variant that relies on a single parameter. Recall that the structural bias refers to the tendency of nodes closer to the training data in the graph to obtain representations that are more aligned with the class centroids, and thus benefit differently from node-level adjustments.

Table 10 reports the calibration results across multiple datasets. The full version consistently achieves lower ECE and better confidence–accuracy alignment than the single-parameter variant. This indicates that modeling the structural bias through separate parameters for training-near and training-distant nodes leads to more appropriate adjustment strengths, thereby yielding improved calibration performance. These results confirm the necessity and effectiveness of the proposed refinement.

Table 10: ECE (%) of each component on all datasets with L/C=20. The best results are highlighted in black.

| Method | GCN | | | | GAT | | | |
|---|---|---|---|---|---|---|---|---|
| | Cora | Citeseer | Pubmed | CoraFull | Cora | Citeseer | Pubmed | CoraFull |
| Only+$\alpha$ | $3.81_{\pm0.61}$ | $3.91_{\pm0.64}$ | $3.94_{\pm0.51}$ | $7.21_{\pm0.52}$ | $3.94_{\pm0.71}$ | $4.88_{\pm0.78}$ | $4.08_{\pm0.72}$ | $6.73_{\pm0.57}$ |
| +$\alpha$ + $\beta$ | $3.35_{\pm0.65}$ | $3.43_{\pm0.58}$ | $3.81_{\pm0.53}$ | $6.96_{\pm0.48}$ | $3.52_{\pm0.74}$ | $4.37_{\pm0.83}$ | $3.78_{\pm0.84}$ | $6.41_{\pm0.63}$ |

### E.7 EVALUATION ON THE OUT-OF-DISTRIBUTION CONDITION.

Table 11: ECE (%) of all methods on all datasets with L/C = 20, using GCN as the base model. Best results are highlighted in black. "ID" and "OOD" denote in-distribution and out-of-distribution, respectively.

| Method | Cora | | Citeseer | |
|---|---|---|---|---|
| | ID | OOD | ID | OOD |
| Uncal. | $13.47_{\pm0.63}$ | $7.15_{\pm0.32}$ | $12.48_{\pm0.71}$ | $10.01_{\pm1.38}$ |
| AU-LS | $4.32_{\pm0.23}$ | $6.75_{\pm0.41}$ | $6.69_{\pm1.76}$ | $8.71_{\pm1.54}$ |
| SCAR | $3.35_{\pm0.65}$ | $4.73_{\pm0.58}$ | $3.43_{\pm0.58}$ | $5.30_{\pm0.91}$ |

To further examine the behavior of our method beyond the in-distribution (ID) setting, we conduct additional experiments on OOD graphs. Following the protocol of Wu et al. (2023), we adopt

Cora-OOD and Citeseer-OOD, where the original citation graph is treated as the in-distribution graph and an OOD graph is generated via a stochastic block model (SBM). Specifically, the SBM graph is sampled with the same number of nodes as the original graph, and node labels are preserved while the edges are randomly rewired according to block-level connectivity probabilities. This setup creates a controlled distribution shift that affects both the graph topology and the smoothness of the label signal.

We evaluate the calibration performance of our method under this OOD setting and compare it with AU-LS, a recent post-hoc calibration baseline designed for graph data. The results are summarized in Table 11. From the Table 11, we have the following observation:

First, the uncalibrated models appear to show improved calibration on OOD graphs. This, however, is largely an artifact: the severe drop in accuracy caused by distribution shift makes the overall accuracy level closer to the confidence scores of under-confident GNNs. Such an apparent "improvement" does not reflect healthy calibration behavior, as it is achieved at the cost of dramatically degraded accuracy. Second, our method consistently improves calibration even under OOD conditions and achieves noticeable gains over AU-LS. This indicates that the proposed approach remains effective under moderate distribution shifts, although the improvement is understandably smaller than in the in-distribution setting.

## F  BROADER IMPACTS

This paper proposes a novel calibration framework for GNNs. For the theoretical analysis of the final-layer weight decay and parameters, providing more explanations and motivations for researching the relationship between the model and confidence in the trustworthy machine learning field. Moreover, the proposed node-level calibration breaks the current reliance on temperature scaling as the sole foundation for graph post-hoc methods, introducing a new calibration framework for graph models. This can offer more choices for the following research and help design more flexible and effective calibration methods.

## G  LLM USAGE STATEMENT

In this work, a large language model (LLM) was used to polish the writing. The LLM assisted in improving clarity and grammar, but all scientific content, interpretations, and conclusions were generated solely by the authors.

