# OpenReview forum: "The Final Layer Holds the Key: A Unified and Efficient GNN Calibration Framework"
_ICLR.cc/2026/Conference — Submitted to ICLR 2026_

### Official Review · Reviewer_LjB2 · 2025-10-31

**Soundness:** 3
**Presentation:** 3
**Contribution:** 3
**Rating:** 6
**Confidence:** 5

**Summary:**

The paper provides a theoretical and practical framework (SCAR) for calibrating GNNs by adjusting final-layer weight decay and introducing a node-level calibration step. It proves that reducing final-layer regularization enlarges class-centroid distances, alleviating under-confidence. A post-hoc correction then nudges node embeddings toward their predicted class centroids, further improving calibration.

**Strengths:**

1. Strong theoretical grounding linking weight decay to confidence underestimation.
2. Dual-level calibration (centroid + node) unifies model-intrinsic and post-hoc methods.
3. Training-free node-level adjustment ensures efficiency and interpretability.
4. Extensive experiments show lower ECE and runtime than prior calibrators

**Weaknesses:**

I haven't seen too much weakness. But I am wondering whether node-level adjustment may amplify overconfidence for misclassified nodes.

**Questions:**

1. How does the method behave when the predicted class centroid is incorrect?
2. Can the framework generalize to heterophilous or dynamic graphs? (Optional)
3. How sensitive are results to the final-layer λ schedule?
4. Could centroid regularization be integrated during training for joint optimization?

**Details Of Ethics Concerns:**

No.

---

> ### Author Response · Authors · 2025-11-15
>
> Thank you for your positive and constructive comments. We address each point in detail below.
>
> >W1&Q1: I am wondering whether node-level adjustment may amplify overconfidence for misclassified nodes. How does the method behave when the predicted class centroid is incorrect?
>
> A1: Thank you for raising these related concerns. We acknowledge that, for a few misclassified nodes, the node-level adjustment may increase their confidence. However, calibration is evaluated at the distribution level rather than per-sample level. For example, when considering all samples whose confidence falls in the range 0.80–0.82, both correctly and incorrectly classified nodes are included, and the goal of calibration is to make their average accuracy match this confidence. Due to the well-known under-confidence of GNNs, the accuracy of nodes in this confidence interval is typically higher than the predicted confidence, so increasing the confidence of all nodes in this bin—including misclassified ones—is necessary to achieve better calibration. Therefore, although local increases in misplaced confidence may occur, this does not lead to systematic overconfidence in aggregate.
>
> In addition, as shown in Eq. (8), the node-level adjustment is applied on top of an enhanced inter-class separability provided by the class-centroid-level calibration. This enlarged margin ensures that the adjustment does not arbitrarily push misclassified nodes toward incorrect centroids. As a result, the refinement remains well-behaved overall and does not amplify overconfidence at the distribution level.
>
> >Q2: Can the framework generalize to heterophilous or dynamic graphs? (Optional)
>
> A2: Thank you for the question. We have already evaluated our framework on heterophilous graphs in Appendix E.3, including both classic heterophilous benchmarks and the large-scale Arxiv-year dataset. The results show that our method generalizes well to heterophilous settings and consistently improves calibration performance. We currently focus on static graphs, and extending the framework to dynamic graphs is an interesting direction for future exploration.
>
> >Q3: How sensitive are results to the final-layer  $\lambda$ schedule?
>
> A3: Thank you for the question. We have already examined the sensitivity of the final-layer $\lambda^{(K)}$ in Appendix E.1. As shown in Figure 5 and Figure 2, varying $\lambda^{(K)}$ leads to smooth and monotonic changes in the reliability diagrams and in the model’s confidence, which is consistent with our theoretical analysis of how the final-layer weight decay affects confidence. This consistent trend is highly beneficial in practice, as it makes the effect of $\lambda^{(K)}$ predictable and easy to search. In fact, because the influence of $\lambda^{(K)}$ on confidence is nearly monotonic, a simple binary search can be used to efficiently identify a suitable value.
>
> >Q4: Could centroid regularization be integrated during training for joint optimization?
>
> A4: Thank you for this insightful suggestion. We agree that integrating centroid regularization into the training process is a promising direction. In this work, however, we deliberately adopt a post-hoc design so that the method remains training-free and can be efficiently applied to any pre-trained GNN without modifying its training pipeline, which is often required in practical deployment. At the same time, our theoretical formulation naturally supports incorporating centroid regularization during training, making joint optimization a meaningful direction for future work.

---

### Official Review · Reviewer_nyom · 2025-10-31

**Soundness:** 3
**Presentation:** 3
**Contribution:** 3
**Rating:** 6
**Confidence:** 4

**Summary:**

This paper proposes SCAR, a nonparametric GNN calibration framework that analyzes miscalibration of GNNs in model perspective. SCAR theoretically shows that reducing a weight decay in the final layer leads to higher class separation, which can mitigate underconfidence of GNNs. It further performs node-level calibration to make the test node’s representation closer to the corresponding label’s centroid.

**Strengths:**

- This paper provides a theoretical connection between underconfidence of GNNs and final layer’s weight decay, which is valuable given the lack of theoretical analysis in GNN calibration literature.
- The proposed method is simple yet effective, avoiding the need to train additional calibration networks as required by many existing methods.
- Extensive experiments shows that SCAR substantially reduces ECE compared to prior baselines, as well as maintaining original classification accuracy of GNNs.

**Weaknesses:**

- The proposed node-level calibration assumes that pushing test nodes toward their predicted class centroids improves confidence, which may not hold under settings such as out-of-distribution (OOD) conditions. For instance, in OOD graphs, pushing test nodes toward centroids learned from training data can degrade calibration.
- If the original GNNs are trained with zero weight decay, the proposed method may be partially inapplicable.
- While SCAR is efficient, it needs to search the optimal configuration over three hyperparameters. Although the authors offer practical heuristics in the appendix (e.g., $\alpha$ should be lower than $\beta$), it is not guaranteed that such heuristics hold universally.

**Questions:**

- Could the authors show the performance of SCAR on OOD graphs?

---

> ### Author Response · Authors · 2025-11-15
>
> Thank you for your positive and constructive comments. We address each point in detail below.
>
> >W1&Q1:The proposed node-level calibration assumes that pushing test nodes toward their predicted class centroids improves confidence, which may not hold under settings such as out-of-distribution (OOD) conditions. For instance, in OOD graphs, pushing test nodes toward centroids learned from training data can degrade calibration. Could the authors show the performance of SCAR on OOD graphs?
>
> A1: Thank you for raising this important point. While our method is primarily designed for in-distribution calibration, we fully agree that examining its behavior under OOD conditions is valuable. Following your suggestion, we conducted additional experiments in OOD graph settings. We use Cora-OOD and Citeseer-OOD following [1], where the original graph is treated as in-distribution and an OOD graph is generated via a stochastic block model. We also include AU-LS as a recent calibration baseline. The results, now reported in Appendix E.7 (marked in blue), show that although the improvement in OOD settings is smaller than in the in-distribution case, our method still achieves consistently better calibration than the baseline by a large margin. This suggests that the proposed method maintains a certain degree of robustness under moderate distribution shifts. We agree that under extreme OOD conditions, the fixed class centroids may become unreliable, and this is a meaningful direction for future investigation.
>
> [1] Wu Q, Chen Y, Yang C, et al. Energy-based out-of-distribution detection for graph neural networks[C]. ICLR, 2023.
>
> |        | Cora-ID       | Cora-OOD  | Citeseer-ID   | Citeseer-OOD  |
> |--------|---------------|---------------|---------------|---------------|
> | Uncal. | 13.47±0.63    | 7.15±0.32     | 12.48±0.71    | 10.01±1.38    |
> | AU-LS  | 4.32±0.23     | 6.75±0.41     | 6.69±1.76     | 8.71±1.54     |
> | SCAR   | **3.35±0.65** | **4.73±0.58** | **3.43±0.58** | **5.30±0.91** |
>
>
> >W2: If the original GNNs are trained with zero weight decay, the proposed method may be partially inapplicable.
>
> A2: Thank you for raising this point. First, we note that training GNNs with zero weight decay is generally known to hurt performance, as multiple studies have shown that an appropriate level of weight decay is crucial for achieving strong generalization in GNNs. When the model’s performance cannot be guaranteed, calibration becomes less meaningful.
>
> Second, if such cases do occur where a GNN trained with zero weight decay still performs best, the proposed node-level calibration remains applicable.
>
> >W3: While SCAR is efficient, it needs to search the optimal configuration over three hyperparameters. Although the authors offer practical heuristics in the appendix (e.g., $\alpha$ should be lower than $\beta$), it is not guaranteed that such heuristics hold universally.
>
> A3: We acknowledge that SCAR involves three hyperparameters, but in practice the tuning cost is modest because of the efficient search strategies we employ. As described in Appendix E.1, the effect of $\lambda$ on model confidence is approximately linear, which allows us to identify a suitable value through a simple binary search. For $\alpha$ and $\beta$, the theoretical relation $\alpha < \beta$ restricts the search space to the upper triangular region, reducing the number of candidate configurations. In addition, these heuristics exhibit stable behavior across different datasets in our experiments, suggesting that they are reasonably robust in practice. We will clarify these points and discuss their empirical motivation in the revised version.

---

> > ### Comment · Reviewer_nyom · 2025-11-20
> > **Response to Authors**
> >
> > Thank you for your detailed response. I will maintain my score toward acceptance.

---

> > > ### Author Response · Authors · 2025-11-20
> > >
> > > Thank you for your time and for the constructive feedback that helps strengthen the paper.

---

### Official Review · Reviewer_kaCQ · 2025-10-31

**Soundness:** 2
**Presentation:** 3
**Contribution:** 3
**Rating:** 4
**Confidence:** 5

**Summary:**

In this paper, the authors conduct a comprehensive analysis of confidence calibration in Graph Neural Networks (GNNs). They first theoretically demonstrate that weight decay applied to the final-layer parameters exacerbates under-confidence by collapsing class centroids toward the origin, thereby reducing class separability. To address this, the authors propose reducing the final-layer weight decay to enhance inter-class distinction and improve confidence calibration at the class-centroid level. Additionally, they introduce a node-level calibration strategy as a fine-grained complement, which encourages each test node to move closer to its predicted class centroid while distancing itself from others in the final-layer representation space, thus improving individual calibration. Finally, they develop a unified theoretical framework that shows model confidence is jointly governed by both class-centroid-level and node-level calibration, underscoring the completeness and coherence of their approach. Extensive experiments demonstrate that the proposed method consistently outperforms state-of-the-art techniques in terms of both effectiveness and efficiency across various datasets and settings.

**Strengths:**

1. The authors are the first to theoretically show that final-layer weight decay aggravates GNN under-confidence, and they mitigate this by reducing the decay.

2. They propose a training-free node-level calibration method as a fine-grained complement to class-centroid-level calibration.

3. They develop a unified theoretical framework showing that both calibration levels jointly govern model confidence, and validate the method’s superiority across diverse settings.

**Weaknesses:**

1. Missing important related work: Given that the paper focuses on confidence calibration, it is concerning that several key papers in the area of uncertainty estimation or calibration for GNNs are not cited or discussed [1-4].

2. Limited baselines: The experimental comparisons would benefit from the inclusion of recent calibration methods [5]

3. Restricted backbone models: The authors only evaluate their method on GCN and GAT. While these are classical models, they are no longer sufficient to represent the landscape of modern GNN architectures. Including additional backbones like GraphSAGE would strengthen the empirical claims and validate the method’s generality.

[1] Uncertainty quantification over graph with conformalized graph neural networks. NeurIPS 2023

[2] Energy-based Epistemic Uncertainty for Graph Neural Networks

[3] Uncertainty Aware Semi-Supervised Learning on Graph Data. NeurIPS 2020

[4] Calibrate Automated Graph Neural Network via Hyperparameter Uncertainty. CIKM 2022

[5] GETS: Ensemble Temperature Scaling for Calibration in Graph Neural Networks. ICLR 2025

**Questions:**

see Weaknesses

---

> ### Author Response · Authors · 2025-11-15
>
> Thank you for your careful review and constructive comments. We provide point-by-point responses to the raised concerns below.
>
> >W1: Missing important related work: Given that the paper focuses on confidence calibration, it is concerning that several key papers in the area of uncertainty estimation or calibration for GNNs are not cited or discussed [1-4].
>
> A1: Thank you for pointing this out. We acknowledge that [1–4] are important works in the broader area of uncertainty estimation for GNNs. Although these papers mainly focus on uncertainty quantification rather than confidence calibration, which explains why they were not included in our initial survey, we agree that they are relevant and should be discussed. We have added all four works to the Related Work section (Appendix A (line 676)) and included a concise analysis of their connection to our work, which is now marked in blue in the updated version.
>
> >W2: Limited baselines: The experimental comparisons would benefit from the inclusion of recent calibration methods [5].
>
> A2: Thank you for the suggestion. Following your advice, we have added comparisons with the recent calibration method [5]. We use the same parameter settings as in the original paper to ensure a fair comparison. The updated results are reported in Tables 1 and 2 of the revised manuscript (marked in blue). The results show that our method achieves an average improvement of 1.6% over GETS across all datasets, settings, and backbones, further supporting the effectiveness of our method in practice.
>
> Backbone: GCN
>
> | L/C=20 | Cora          | Citeseer      | Pubmed        | Cora-Full     |
> |--------|---------------|---------------|---------------|---------------|
> | GETS   | 3.83±0.41     | 6.72±0.78     | 3.93±0.37     | 7.01±0.85     |
> | SCAR   | **3.35±0.53** | **3.43±0.58** | **3.81±0.47** | **6.96±0.48** |
>
> | L/C=40 | Cora          | Citeseer      | Pubmed        | Cora-Full     |
> |--------|---------------|---------------|---------------|---------------|
> | GETS   | 3.72±0.39     | 6.01±0.63     | 3.72±0.29     | 5.98±0.74     |
> | SCAR   | **2.67±0.33** | **4.17±0.45** | **3.16±0.52** | **5.61±0.69** |
>
> | L/C=60 | Cora          | Citeseer      | Pubmed        | Cora-Full     |
> |--------|---------------|---------------|---------------|---------------|
> | GETS   | 3.26±0.67     | 7.84±0.66     | 3.03±0.26     | 6.83±0.41     |
> | SCAR   | **2.60±0.62** | **4.72±0.69** | **2.67±0.63** | **6.31±0.69** |
>
> Backbone: GAT
>
> | L/C=20 | Cora          | Citeseer      | Pubmed        | Cora-Full     |
> |--------|---------------|---------------|---------------|---------------|
> | GETS   | 4.53±0.48     | 5.85±0.33     | 4.16±0.37     | 6.79±0.71     |
> | SCAR   | **3.52±0.74** | **4.37±0.83** | **3.78±0.84** | **6.41±0.63** |
>
> | L/C=40 | Cora          | Citeseer      | Pubmed        | Cora-Full     |
> |--------|---------------|---------------|---------------|---------------|
> | GETS   | 4.01±0.68     | 5.43±0.47     | 3.34±0.44     | 5.95±0.54     |
> | SCAR   | **3.52±0.74** | **3.54±0.65** | **3.02±0.30** | **5.52±0.46** |
>
> | L/C=60 | Cora          | Citeseer      | Pubmed        | Cora-Full     |
> |--------|---------------|---------------|---------------|---------------|
> | GETS   | 3.16±0.58     | 5.39±0.76     | 2.96±0.61     | 5.21±0.69     |
> | SCAR   | **2.59±0.43** | **4.24±0.35** | **2.80±0.85** | **4.75±0.64** |
>
>
>
> >W3: Restricted backbone models: The authors only evaluate their method on GCN and GAT. While these are classical models, they are no longer sufficient to represent the landscape of modern GNN architectures. Including additional backbones like GraphSAGE would strengthen the empirical claims and validate the method’s generality.
>
> A3: Thank you for the insightful suggestion. We would like to clarify that Appendix E.4 already includes experiments using additional representative GNN architectures such as GCNII and FAGCN. Following your recommendation, we have further added experiments with GraphSAGE as another backbone, and the results have been incorporated into Appendix E.4 in the revised version. Across all these architectures, our method consistently improves calibration performance, which supports its generality beyond GCN and GAT.
>
> |          | Cora          | Citeseer      | Pubmed        | Chameleon     | Squirrel      |
> |-----------|-----------|---------|--------------|---------------|---------------|
> | FAGCN          | 15.94±0.17    | 20.56±4.51    | 5.83±0.63     | 15.83±0.63    | 10.36±0.24    |
> | SCAR-FAGCN     | **4.57±0.60** | **4.92±1.42** | **3.74±0.71** | **6.96±0.84** | **7.85±0.43** |
> | GCNII          | 35.51±0.56    | 35.30±1.11    | 5.72±1.77     | 18.13±0.26    | 17.79±0.61    |
> | SCAR-GCNII     | **9.81±0.48** | **8.74±0.51** | **3.98±0.88** | **7.03±0.35** | **8.93±0.37** |
> | GraphSAGE      | 9.89±0.78     | 10.08±0.88    | 7.38±0.65     | 13.27±0.81    | 11.14±0.48    |
> | SCAR-GraphSAGE | **4.88±0.71** | **4.32±0.73** | **5.86±0.46** | **6.74±0.86** | **7.17±0.51** |

---

> > ### Author Response · Authors · 2025-11-26
> >
> > Dear Reviewer kaCQ,
> >
> > Thank you once again for your valuable and constructive feedback on our submission. We want to kindly confirm if we have adequately addressed all your concerns. Please do not hesitate to let us know if there are any remaining questions or points that require further clarification.
> >
> > Sincerely,
> >
> > Authors of Submission 12439

---

### Official Review · Reviewer_BypW · 2025-11-03

**Soundness:** 3
**Presentation:** 3
**Contribution:** 3
**Rating:** 8
**Confidence:** 3

**Summary:**

This paper tackles the problem of confidence miscalibration in graph neural networks. The authors observe that GNN confidence is influenced by two factors in the final layer, namely Class-Centroid-Level Calibration and Node-Level Calibration. Building on this insight, they propose the SCAR framework, which unifies these two calibration components into a single theoretical framework, enabling more effective confidence calibration in GNNs.

**Strengths:**

1. This is the paper's most significant strength. It moves beyond heuristic-based calibration by providing a rigorous theoretical analysis.

2. The proposed SCAR method consistently outperforms a wide range of strong baselines across multiple datasets.

**Weaknesses:**

1. The node-level calibration is refined in Eq. 10 to account for the structural bias of GNNs (nodes closer to training data get more similar representations). While this is a thoughtful addition, its evaluation is limited. An ablation study showing the performance gain of using two parameters $\alpha$ and $\beta$ over a single one would have strengthened this claim.

2. The details of the high-order neighbors of the training node is not well specified.

3. Sensitivity analysis on hyper-parameter $\lambda^{(k)}$ is not provided.

**Questions:**

see weakness.

---

> ### Author Response · Authors · 2025-11-15
>
> We thank the reviewer for the positive assessment and the constructive suggestions. Our point-by-point responses are provided below.
>
> >W1: The node-level calibration is refined in Eq. 10 to account for the structural bias of GNNs (nodes closer to training data get more similar representations). While this is a thoughtful addition, its evaluation is limited. An ablation study showing the performance gain of using two parameters $\alpha$ and $\beta$ over a single one would have strengthened this claim.
>
> A1: We thank the reviewer for this constructive suggestion. Following your advice, we have conducted an additional ablation study comparing the proposed node-level calibration using both parameters ($\alpha$ and $\beta$) with a simplified version that employs only a single parameter.
>
> The results (shown in Table 10 and detailed in Appendix E.6, marked in blue) demonstrate that the node-level calibration that explicitly accounts for the structural bias of GNNs further improves calibration performance compared to using a single parameter. This supports the effectiveness of the refined design.
>
> | GCN               | Cora      | Citeseer  | Pubmed    | Cora-Full |
> |-------------------|-----------|-----------|-----------|-----------|
> | Only+ $\alpha$    | 3.81±0.61 | 3.91±0.64 | 3.94±0.51 | 7.21±0.52 |
> | +$\alpha$+$\beta$ | 3.35±0.65 | 3.43±0.58 | 3.81±0.53 | 6.96±0.48 |
>
> | GAT               | Cora       | Citeseer  | Pubmed    | Cora-Full |
> |-------------------|------------|-----------|-----------|-----------|
> | Only+ $\alpha$    | 3.940±0.71 | 4.88±0.78 | 4.08±0.72 | 6.73±0.57 |
> | +$\alpha$+$\beta$ | 3.52±0.74  | 4.37±0.83 | 3.78±0.84 | 6.41±0.63 |
>
> >W2: The details of the high-order neighbors of the training node is not well specified.
>
> A2: Thank you for pointing this out. In our paper, “high-order neighbors” refers to second-order and higher-order neighbors of the training nodes, i.e., nodes that are not directly connected to a training node but can be reached only through multi-hop paths. We have clarified this definition in the revised manuscript at line 304 (marked in blue).
>
> >W3: Sensitivity analysis on hyper-parameter $\lambda^{(K)}$ is not provided.
>
> A3: Thank you for pointing this out. We have already provided a sensitivity analysis of the hyper-parameter $\lambda^{(K)}$ in Appendix E.1 (Figure 5), where we visualize the reliability diagrams under different values of $\lambda^{(K)}$. The results show a clear and smooth trend: as $\lambda^{(K)}$ decreases, the model’s confidence gradually shifts from under-confident to more confident, which is fully consistent with our theoretical analysis of how the final-layer weight decay influences confidence. We will highlight this more explicitly in the revised version.

---

> > ### Comment · Reviewer_BypW · 2025-11-24
> > **comment**
> >
> > Thanks for the rebuttal. The responses have addressed my concerns. I will keep my score.

---

> > > ### Author Response · Authors · 2025-11-25
> > >
> > > Thank you for your time and for maintaining your positive score. We appreciate your supportive evaluation.

---

### Meta-Review · Area_Chair_Y7Fd · 2026-01-07

**Summary:**

This paper proposes SCAR, a unified calibration framework for GNNs that explains miscalibration through class-centroid-level and node-level effects. It shows that reducing final-layer weight decay improves class separation and mitigates under-confidence, while node-level calibration provides fine-grained adjustments by pulling each test node toward its predicted class centroid.

While most reviewers lean toward acceptance, I share the concerns raised by Reviewer kaCQ regarding the empirical evaluation. In particular, I find the experimental campaign insufficient when compared to prior literature (e.g., GETS). The authors initially evaluate the method on only a few small-scale homophilic datasets (Cora, Citeseer, and Pubmed), following the setup of [1]. Although additional datasets are included in the appendix, performance results for baseline methods are missing, making it difficult to assess the advantage of SCAR. In addition, the results authors provide for GETS during the rebuttal substantially diverges from those reported in the original manuscript.

All that being said, I also do not believe the work stands on its own as a theory paper. Several statements/derivations lack sufficient rigor. For instance, Theorem 3.1 aims to characterize how the probabilities $s_{v,i}$ evolve after one epoch. However, the derivation implicitly assumes that the update of $W^{(K)}$ depends only on $\mathcal{L}_v$, effectively ignoring the loss contributions $\mathcal{L}_u$ from other training nodes ($u \neq v$). Moreover, a quick inspection of the proofs reveals clear mistakes - for example, in Eq. (13):
$$
\frac{\partial \mathcal{L}\_v}{\partial \mathbf{W}^{(K)}\_{:,i}}=\sum\_{j=1}^c
\left(
\frac{\partial \mathcal{L}\_v}{\partial s\_{v,j}}
\frac{\partial s\_{v,j}}{\partial z\_{v,i}}
\right)
\frac{\partial z\_{v,i}}{\partial \mathbf{W}^{(K)}\_{:,i}}
+
\frac{\partial \mathcal{L}\_v}{\mathbf{W}^{(K)}\_{:,i}}
$$

Clearly, it is missing a $\partial$ in the right-most denominator. Even after fixing this, the equation as written would imply that the summation term is zero. Also, it is also worth mentioning that $\mathcal{L}_v$ (Eq. (4)) essentially reduces to logistic regression (when $c=2$) with a Gaussian prior for a linear model, given that all $W^{(k)}$ for $k < K$ are treated as fixed. For this objective, it is well known that there is no closed-form solution, which raises a red flag regarding Theorem 3.2.

Overall, I believe the paper requires substantial revisions and is not yet ready for acceptance.

[1] Be Confident! Towards Trustworthy Graph Neural Networks via Confidence Calibration, NeurIPS 2021.

**Reviewer Concerns:**

Reviewer ``BypW`` raised concerns regarding the lack of ablation studies and sensitivity analyses on hyperparameters. In response, the authors provided additional experimental results. The reviewer acknowledged the rebuttal and indicated that they would maintain their initial positive evaluation (score: 8).

Reviewer ``kaCQ`` raised concerns about missing related work, limited calibration baselines, and the restricted set of backbone models. While the authors added results using GETS, the reported numbers differ substantially from those in the original papers, and it is unclear what accounts for this discrepancy. Furthermore, in the additional experiments with GraphSAGE (and other GNNs), the authors report results only for the uncalibrated model versus their proposed calibration method, omitting comparisons with existing calibration baselines. Finally, the evaluation relies heavily on homophilic datasets, which may limit the generality of the conclusions.

Reviewer ``nyom`` raised concerns regarding the performance on out-of-distribution (OOD) data, the method’s reliance on models trained with weight decay (raising questions about applicability), and the computational overhead associated with model selection due to additional hyperparameters. Among these points, I believe the reliance on weight decay is only partially addressed. Overall, I disagree with the authors’ claim that an “appropriate level of weight decay is crucial for achieving strong generalization in GNNs,” as there exist multiple regularization strategies capable of yielding strong generalization performance.

Reviewer ``LjB2`` raised concerns regarding sensitivity to the lambda scheduling strategy and the lack of performance evaluation on heterophilic graphs. In my view, these issues remain largely unaddressed. In particular, the results presented in Appendix E.1 compare only the uncalibrated model with the proposed method, again omitting comparisons against existing calibration baselines on heterophilic datasets.

**Reviewer Scores:**

Two reviewers responded during the discussion phase, indicating that they would maintain their initial level of support: Reviewer ``BypW`` (score: 8) and Reviewer ``nyom`` (score: 6). I believe that Reviewers ``kaCQ`` and ``LjB2`` are unlikely to increase their scores, as the additional experiments provided do not sufficiently address their core concerns.

---

### Decision · Program_Chairs · 2026-01-26

Reject